# Intrinsic factors responsible for brittle versus ductile nature of refractory high-entropy alloys

Tomohito Tsuru [1,2] ✉, Shu Han [3], Shutaro Matsuura[3], Zhenghao Chen [3], Kyosuke Kishida [2,3] ✉, Ivan Iobzenko[1], Satish I. Rao[4], Christopher Woodward[5], Easo P. George [6,7] ✉ & Haruyuki Inui [2,3] ✉

Refractory high-entropy alloys (RHEAs) are of interest for ultrahigh-temperature applications. To overcome their drawbacks − low-temperature brittleness and poor creep strength at high temperatures − improved fundamental understanding is needed. Using experiments, theory, and modeling, we investigated prototypical body-centered cubic (BCC) RHEAs, TiZrHfNbTa and VNbMoTaW. The former is compressible to 77 K, whereas the latter is not below 298 K. Hexagonal close-packed (HCP) elements in TiZrHfNbTa lower its dislocation core energy, increase lattice distortion, and lower its shear modulus relative to VNbMoTaW whose elements are all BCC. Screw dislocations dominate TiZrHfNbTa plasticity, but equal numbers of edges and screws exist in VNbTaMoW. Dislocation cores are compact in VNbTaMoW and extended in TiZrHfNbTa, and different macroscopic slip planes are activated in the two RHEAs, which we attribute to the concentration of HCP elements. Our findings demonstrate how ductility and strength can be controlled through the ratio of HCP to BCC elements in RHEAs.

Ni-based superalloys have evolved into highly engineered materials that are the current state-of-the-art in metallic alloys for structural applications at high temperatures in oxidizing environments[1]. The most advanced of these alloys, e.g., CMSX-4, have a face-centered cubic (FCC) matrix strengthened by a high volume-fraction of $L1_2$-structured $Ni_3Al$-type precipitates arrayed in a brick-and-mortar pattern. One of their most demanding applications is turbine blades for the high-pressure section of modern jet engines in which gas temperatures can exceed the melting temperature of superalloys while the alloys themselves are kept at lower temperature with the help of insulating thermal barrier coatings on blade surfaces and integral cooling channels within the blades[2]. Further increases in engine and power-plant efficiencies (essential for decreasing their carbon footprint) will require increases in operating temperature. However, given how close current operating conditions are to the inherent limits of existing materials, new classes of alloys with much higher melting temperatures will be needed.

One such class comprises the RHEAs which, because of their high melting points, have been proposed as candidates for ultrahigh-temperature applications[3–6]. However, a single-phase RHEA, TiZrHfNbTa, was recently shown to be significantly weaker in creep than CMSX-4 despite the former's higher melting point[7]. The weakness was ascribed to two factors: faster diffusion in the BCC matrix of the RHEA relative to the FCC matrix of CMSX-4 and a lack of precipitates.

[1]Nuclear Science and Engineering Center, Japan Atomic Energy Agency, 2-4 Shirakata, Tokai-mura, Ibaraki 319-1195, Japan. [2]Center for Elements Strategy Initiative for Structural Materials (ESISM), Kyoto University, Sakyo-ku, Kyoto 606-8501, Japan. [3]Department of Materials Science and Engineering, Kyoto University, Sakyo-ku, Kyoto 606-8501, Japan. [4]Department of Mechanical Engineering, Johns Hopkins University, Baltimore, MD 21218, USA. [5]Materials and Manufacturing Directorate, Air Force Research Laboratory (retired), Wright Patterson Air Force Base, Dayton, OH 45433-7817, USA. [6]Department of Materials Science and Engineering, University of Tennessee, Knoxville, TN 37996, USA. [7]Institute for Materials, Ruhr University Bochum, 44801 Bochum, Germany. ✉e-mail: tsuru.tomohito@jaea.go.jp; kishida.kyosuke.6w@kyoto-u.ac.jp; egeorge@utk.edu; inui.haruyuki.3z@kyoto-u.ac.jp

Additional research is therefore needed to enhance the creep strength of TiZrHfNbTa, for example, by the introduction of suitable strengthening precipitates. Meanwhile, RHEAs with higher melting points must also be investigated since diffusion will be slower in such materials. One such RHEA is VNbMoTaW[8], whose melting point is ~500 °C higher than TiZrHfNbTa[7,9,10]. However, it has a major disadvantage: lack of ductility at room temperature even in compression[8]. In contrast, TiZrHfNbTa exhibits ductility at room temperature both in compression[11] and in tension (~8%)[12], albeit not as much as the highly ductile and fracture-resistant FCC HEAs[13–16].

Structural materials, even those targeting primarily high-temperature applications, need some minimal ductility at room temperature to withstand routine handling, accidental drops, thermal shocks, etc. The dilemma we face is that TiZrHfNbTa possesses tensile ductility[12,17] but poor creep strength[7], whereas VNbMoTaW while potentially having sufficient creep strength (although that remains to be determined), is dead brittle.

In the present work, we utilize experiments, theory, and modeling to investigate the above two model systems: TiZrHfNbTa (ductile) and VNbMoTaW (brittle), with the overarching goal of developing a fundamental understanding of the key factors that control their ductility and strength to help guide the design of new RHEAs for next-generation high-temperature structural applications.

## Results

### Temperature dependence of yield stress and activation enthalpy

In the present study, BCC single-phase VNbMoTaW and TiZrNbHfTa equiatomic alloys were prepared. From compression tests on polycrystalline VNbMoTaW and TiZrNbHfTa, the temperature dependence of yield stress was obtained. Examples of compressive stress-strain curves at temperatures below and above the respective brittle-to-ductile transition temperatures (BDTTs) are shown in Fig. 1a, b. Consistent with earlier reports[8,11,12], VNbMoTaW is brittle while TiZrNbHfTa is ductile at room temperature. The temperature dependences of yield

stress are plotted in Fig. 1c. Over almost the entire temperature range, VNbMoTaW is stronger than TiZrNbHfTa, consistent with previous results[3]. In VNbMoTaW, room temperature was the lowest temperature at which plastic flow could be observed, whereas TiZrNbHfTa could be plastically compressed down to 77 K. Although there is a big difference in the onset temperature for plastic flow, the temperature dependence of yield stress is qualitatively similar in the two alloys as well as other BCC metals[18–20]: yield stress increases rapidly at low temperatures, and it stays at an almost constant value, 'athermal' stress ($\sigma_\mu$) above a critical or 'athermal' temperature ($T_{TA}$). The athermal stress and temperature are determined from Fig. 1c to be 830 MPa and 973 K for VNbMoTaW and 798 MPa and 700 K for TiZrNbHfTa. From these values, the effective yield stress at 0 K [$\sigma_{th}(0)$] for the two alloys can be estimated by fitting the yield stress-temperature curves in Fig. 1c to the equation in Supplementary Table 1. The extrapolated curves in Fig. 1c suggest a steeper rise in the yield strength of TiZrNbHfTa compared to that of VNbMoTaW at cryogenic temperatures making the former's $\sigma_{th}(0)$ higher than that of the latter (Supplementary Table 1). Given that the macroscopic yield stress is expressed by the orthogonal transformation of the critical resolved shear stress (CRSS), i.e., $\sigma_{th} \propto \tau_{CRSS}$, yield stresses can be normalized by the 0 K shear modulus ($G$), and test temperatures ($T$) by the melting temperature ($T_m$), and those results are shown in Fig. 1d. In this study, $G$ along <111> on {110} plane (which is the orientation used for the CRSS) was determined at 0 K using first-principles calculations: 74.0 for VNbMoTaW and 28.8 GPa for TiZrNbHfTa (more on this later); melting temperatures ($T_m$) were taken from earlier work[10]: 2682 and 2155 K. In the normalized plots, TiZrNbHfTa is found to be stronger than VNbMoTaW at all homologous temperatures, in contrast to the absolute values in Fig. 1c.

In many brittle materials, the brittle-to-ductile transition (BDT) occurs in a relatively narrow temperature range called the BDTT[21–23]. Such a BDT is triggered by a rapid increase in thermally activated dislocation activity and thus is strain-rate dependent[21,24]. The activation

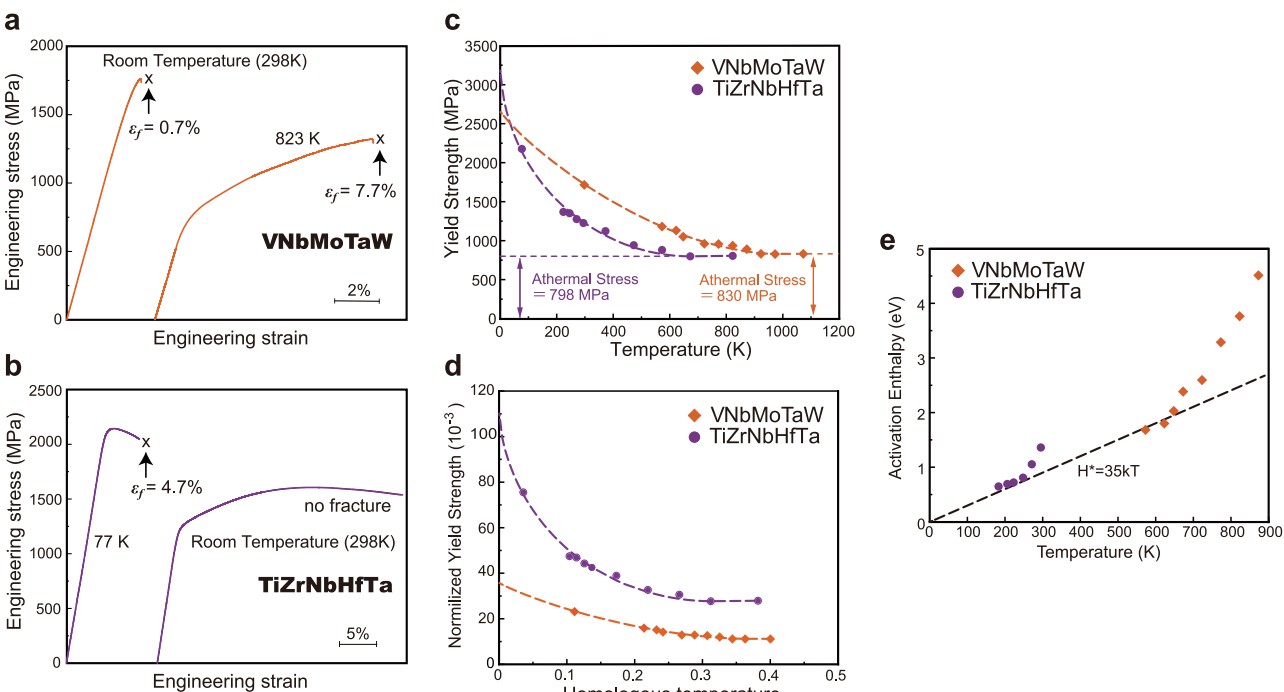

**Fig. 1 | Compressive mechanical properties of equiatomic VNbMoTaW and TiZrNbHfTa RHEAs. a, b** Stress-strain curves for VNbMoTaW (**a**) and TiZrNbHfTa (**b**) compressively deformed at temperatures below and above their respective BDTTs. Fracture occurred in all cases, except in TiZrNbHfTa at room temperature; the corresponding fracture strains ($\varepsilon_f$) are labeled in the plots. **c, d** Temperature dependence of yield strength (**c**), shear modulus-normalized yield strength as a function of homologous temperature ($T/T_m$) (**d**), and temperature dependence of activation enthalpy for dislocation motion (**e**).

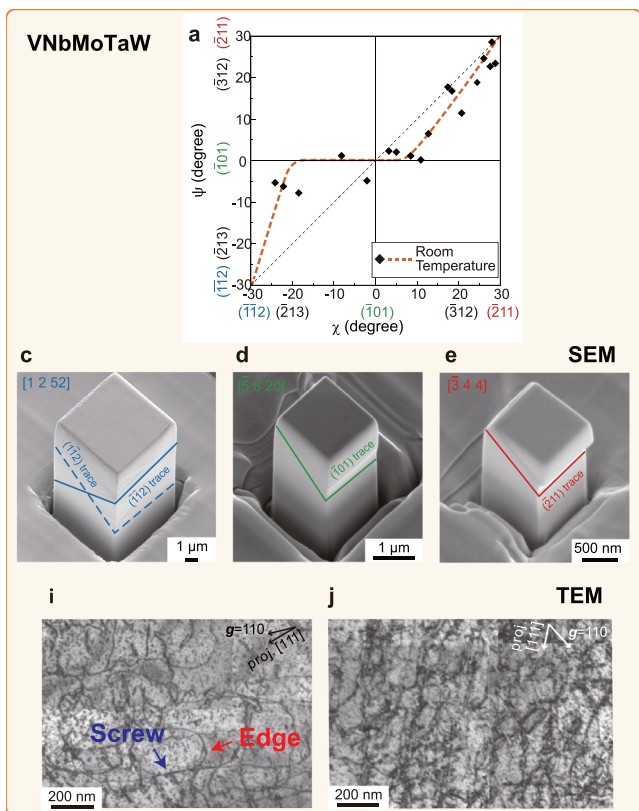

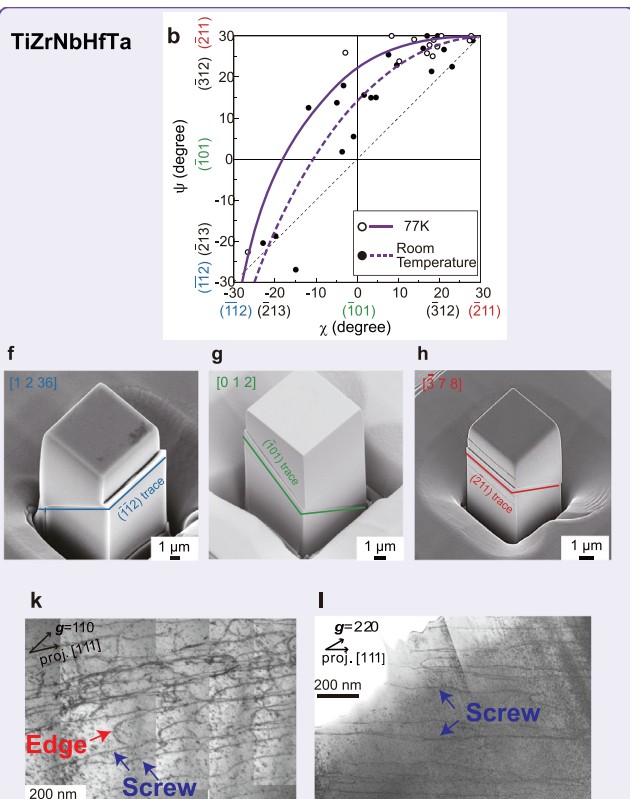

**Fig. 2 | Slip traces and dislocation structures in VNbMoTaW and TiZrNbHfTa after compression to 2 ~ 3% strain. a, b** ψ-χ plots for VNbMoTaW (**a**) at room temperature and TiZrNbHfTa (**b**) at room temperature and 77 K (ψ-χ are defined in Supplementary Fig. 2). **c–e** Typical deformed micropillars of VNbMoTaW after loading along [1 2 52](**c**), [5̄ 8 20](**d**), and [3̄ 4 4](**e**). **f–h** Typical deformed micropillars of TiZrNbHfTa after loading along [1 2 36](**f**), [0 1 2](**g**), [3̄ 7 8](**h**). **i, j** Bright-field TEM images of typical deformation microstructure in VNbMoTaW afterslip on (2̄11) (**i**) and (1̄ 0 1) (**j**) at room temperature. **k, l** Bright-field TEM images of typical deformation microstructure in TiZrNbHfTa afterslip on (2̄11) at room temperature (**k**) and 77 K (**l**). The thin foil for (**k**) was cut parallel to the (2̄11) macroscopic slip plane of the micropillar specimen deformed along the twinning direction. For comparison, (**l**) was cut parallel to the same plane from a deformed bulk polycrystal.

enthalpy for deformation ($H^*$) in BCC metals at low temperatures follows an empirical expression of type $H^* = AkT$, where $A$ is a constant in the range 25–35[21,24]. In the present study, $H^*$ was evaluated from strain-rate jump tests (Supplementary Fig. 1), and the results are plotted in Fig. 1e, which shows that $A$ is 35 for both VNbMoTaW and TiZrNbHfTa. This empirical expression is valid for thermally activated processes involving kink-pair nucleation and/or migration. Therefore, when a deviation from this empirical expression occurs, it implies that kink-pair activation is no longer rate-controlling above that temperature because of a rapid increase in the mobile dislocation density[19,25–28]. The temperature at which the deviation occurs may be taken as the BDTT[27]. Using such a procedure, the BDTTs in Fig. 1e are estimated to be 627 K for VNbMoTaW and 247 K for TiZrNbHfTa (above and below room temperature, respectively). In terms of homologous temperatures, the BDTTs are 0.213 and 0.098 $T_m$ for VNbMoTaW and TiZrNbHfTa. The BDTT is much higher for VNbMoTaW than for TiZrNbHfTa, consistent with previously reported tensile ductilities of the two RHEAs[8,11,29]. Thus, the low tensile ductility of VNbMoTaW at room temperature (which is well below its BDTT) arises from low dislocation mobility.

### Deformation behaviors at room temperature and 77 K

In conventional BCC alloys, macroscopically observed slip planes depend on the loading axis because <111> dislocations frequently cross-slip among {110} and {112} planes. The selection of slip planes is usually described using the ψ-χ plot as explained in Supplementary Fig. 2a and Methods. For VNbMoTaW and TiZrNbHfTa, the ψ-χ plots are summarized in Fig. 2a, b, which indicate that (1̄01) slip is preferred in VNbMoTaW whereas (2̄11) slip is preferred in TiZrNbHfTa with the latter tendency becoming more pronounced at 77 K.

To verify the aforementioned slip on (1̄01), (2̄11), and (1̄1̄2) planes, single crystal micropillars were compressed along different loading axes (χ = 0 and ±30°). Selected stress-strain curves are shown in Supplementary Figs. 3a–c for VNbMoTaW and in Supplementary Figs. 3d–f for TiZrNbHfTa. Micrographs taken with a scanning electron microscope (SEM) along a viewing direction inclined 30° to the loading-axis direction are shown in Fig. 2c–e for VNbMoTaW and Fig. 2f–h for TiZrNbHfTa. Unlike the slip lines in bulk polycrystals (Supplementary Fig. 2c), the slip traces in all micropillar specimens are fairly straight. For both RHEAs, the operative slip planes are (2̄11) and (1̄1̄2) for the twinning and anti-twinning shear directions. (1̄1̄2) is also observed for the anti-twinning shear in VNbMoTaW as shown in Supplementary Fig. 3c. For shear at χ = 0° (where (1̄01) is the maximum resolved shear stress plane), (1̄01) is identified as the operative slip plane in VNbMoTaW, whereas in TiZrNbHfTa the operative slip plane deviates from (1̄01) by about +12°. The results of slip trace analysis on compressed micropillars are identical to those obtained on bulk polycrystals as plotted in Fig. 2a, b, confirming that (1̄01) slip is preferred in VNbMoTaW, whereas (2̄11) slip is preferred in TiZrNbHfTa.

Figure 2i, j show dislocation structures in VNbMoTaW deformed to 2-3% plastic strain at room temperature in two directions (χ = +30° and 0°). Unlike in most BCC alloys, screw and edge dislocation segments are present almost equally here regardless of the operative slip plane ((2̄11) or (1̄01)). In contrast, screw dislocations dominate the microstructure of TiZrNbHfTa at both room temperature and 77 K

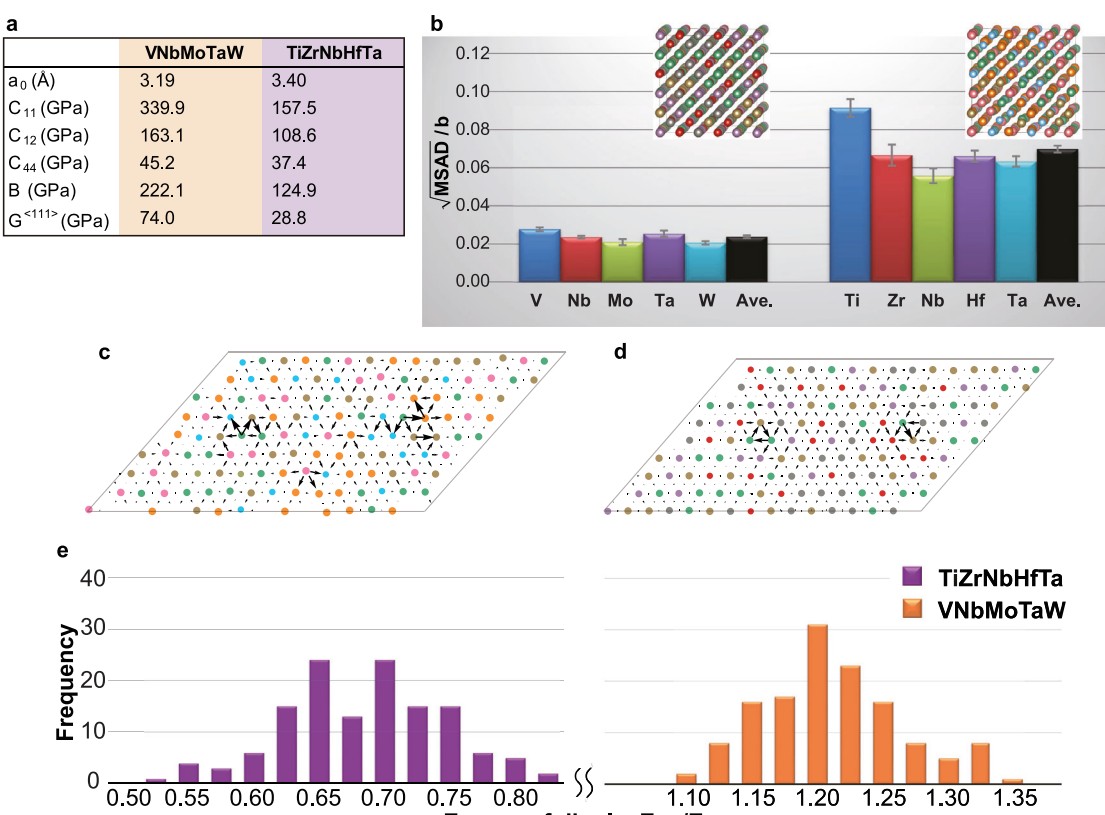

**Fig. 3 | Fundamental properties, assessed at 0 K, associated with the mechanical properties of VNbMoTaW and TiZrNbHfTa. a** Lattice and elastic constants of VNbMoTaW and TiZrNbHfTa with ideally random configuration. **b** Square root of the MSAD values of each constitutive element, and the average of all elements, normalized by the Burgers vector (the error bars represent standard deviations). Inserted atomic images are typical examples of the relaxed configuration (See Supplementary Fig. 4). **c, d** Typical configuration of a screw dislocation dipole in TiZrNbHfTa (**c**) and VNbMoTaW (**d**), where the dislocation core is visualized using the differential displacement vector. **e** Frequency distribution of the core energy normalized by the energy calculated by the elasticity theory for 135 dipole configurations in TiZrNbHfTa and VNbMoTaW (note the break in the energy scale, horizontal axis).

(Fig. 2k, l), albeit with the number of edge segments increasing with temperature from 77 K to room temperature, as has also been observed previously[30,31]. These observations suggest that the deformation of TiZrNbHfTa is controlled by the motion of screw dislocations, while plasticity in VNbMoTaW is affected by both edge and screw dislocations.

## Yield strength modeling

From yield strength modeling based on screw dislocation strengthening (see "Methods" section), the room-temperature yield strengths of TiZrNbHfTa and VNbMoTaW are determined to be 800 and 1242 MPa, respectively. Similar calculations based on edge dislocation strengthening give values of 698 and 1210 MPa for TiZrNbHfTa and VNbMoTaW. These predictions are in line with our experimentally measured yield strengths, but approximately 25–30% lower in absolute terms. Taking account of the solute-screw dislocation interaction energies in the strengthening model, Supplementary Tables 2 and 3, and the fact that the average vacancy and self-interstitial formation energies are higher in VNbMoTaW[32], screw dislocation strengthening is predicted to be more significant for VNbMoTaW than for TiZrNbHfTa. Additionally, due to the significantly higher shear modulus of VNbMoTaW, edge dislocation strengthening is also predicted to be more significant for VNbMoTaW. In other words, regardless of whether a screw or edge dislocation strength model is used, the predicted yield strength of TiZrNbHfTa is lower than that of VNbMoTaW, consistent with experiments (Fig. 1c). However, the magnitudes of screw and edge dislocation strengthening are predicted to be similar for VNbMoTaW, which agree qualitatively with the observed post-deformation

dislocation structures, namely that edge and screw segments are present in roughly equal numbers in VNbMoTaW.

## Elastic constants, local displacements, and dislocation core features from first-principles calculations

Average lattice and elastic constants are calculated using a standard density functional theory approach (see "Methods" section) and are summarized in Fig. 3a. The elastic constants of TiZrNbHfTa are much smaller than those of VNbMoTaW (note that $G$ is the shear modulus along <111>). The modulus-normalized yield stress of TiZrNbHfTa is relatively large due to this low shear modulus (Fig. 1d).

Lattice distortion is an important feature of HEAs[33]. A simple $\delta$ parameter associated with the difference in the atomic radius of the constituent elements has been used to estimate the magnitude of lattice distortion[34]. The atomic distortion and the $\delta$ parameters for the two RHEAs of this study are shown in Supplementary Fig. 4. The $\delta$ parameter is, however, not sufficient to describe the displacement of each constituent element. Therefore, taking advantage of full atomistic simulation, the mean square atomic displacement (MSAD)[35] was calculated instead. Figure 3b and Supplementary Table 4 shows the MSAD values of each constituent element, along with their average, normalized by the magnitude of the Burgers vector ($a_0/2$<112>). Unlike the $\delta$ parameter, there is a clear difference in the normalized MSAD values of the two RHEAs. The MSAD of VNbMoTaW is close to that of FCC HEAs[36], while that of TiZrNbHfTa is quite large, exceeding 6% of the Burgers vector. This substantial lattice distortion (large MSAD) contributes to not only the volumetric strain but also the shear strain

components as shown in Supplementary Fig. 5, resulting in the higher modulus-normalized yield stress of TiZrNbHfTa in Fig. 1d.

Dislocation core structures of VNbMoTaW and TiZrNbHfTa were also investigated. A screw dislocation dipole was inserted into 135 equivalent positions, and its formation energy was calculated for each possible configuration. Typical examples of dislocation core structures in the two RHEAs are shown in Fig. 3c, d, where the dislocation core was identified using differential displacement vectors[37]. The dislocation in VNbMoTaW has a compact core, as commonly seen in pure BCC metals[38]. On the other hand, the screw dislocation in TiZrNbHfTa is not compact. This extended core is due to its tendency for phase instability (more on this later), which originates in the electronic structure of the constituent elements. The energies of dislocation dipoles in RHEAs may vary depending on the local atomic arrangement around the dislocation core. Figure 3e shows the frequency distribution for 135 dipole configurations in VNbMoTaW and TiZrNbHfTa, normalized by the dipole energy based on elasticity theory (see "Methods" section). Core energy before normalization is given in Supplementary Table 5. As expected, the core energy is distributed over a wide energy range. More importantly, however, there is a significant difference between the average core energies of the two RHEAs. The energy of the dislocation core in TiZrNbHfTa is much lower than that of VNbMoTaW, indicating that a dislocation can be introduced more easily in TiZrNbHfTa. Since plastic deformability (ductility) depends partly on the ease of dislocation nucleation and partly on the ease of dislocation motion, our results offer a plausible explanation for why TiZrNbHfTa should have better ductility and lower BDTT relative to VNbMoTaW.

Based on the above results, we hypothesized that the higher concentration of HCP elements in TiZrNbHfTa than in VNbMoTaW is what causes a significant change in lattice distortion, dislocation core structure, and slip behavior and that the same may be the case in other BCC RHEAs. A theoretical ductility criterion in BCC alloys has been proposed using a virtual crystal approximation (VCA) scheme, where affine deformation was applied to evaluate the relationship between mechanical response and electronic structure[39]. In the present study, virtual Nb-Zr alloys were constructed with different compositions: pure Nb, Nb67Zr33, and Nb50Zr50, to test our hypothesis regarding the effect of increased HCP-element concentration. Dislocation trajectories were investigated for these compositions by first-principles calculations with the VCA scheme (see "Methods" section). The Peierls energy surface is evaluated as shown in Fig. 4, in which a screw dislocation moves within the triangle surrounded by the easy-, hard-, and split-cores (Fig. 4a). It can be seen from Fig. 4b that the minimum energy path is not consistent with the commonly occurring ($\bar{1}01$) slip plane but leans slightly toward the ($\bar{2}11$) plane. The tilt depends on the element type, namely the electronic structure of $d$-bands[38]. VCA allows us to predict the effect of the change in $d$-bands on the minimum energy path for screw dislocation motion. Interestingly, the minimum energy path coincides with the ($\bar{2}11$) plane when the concentration of Zr reaches 50 at%. Admittedly, the binary alloys considered in the above treatment are a crude approximation of the real quinary RHEAs, made necessary by limitations of our computing capability. Further research employing more realistic compositions are needed for a more comprehensive understanding.

The phonon dispersion and density of states of these virtual alloys are shown in Fig. 4c. The phonon modes along the [ξξ0] and [ξξξ] correspond to the BCC-HCP and BCC-ω transitions, respectively. These transformations induce the local atomic shuffling associated with in-plane displacement along <1$\bar{1}$0>{110} for BCC-HCP and <111>{$\bar{2}$11} for BCC-ω transitions[40–42]. This local shuffling leads to the large lattice distortion in TiZrNbHfTa, while the electronic structure of HCP elements contributes to its low shear modulus; the latter effect has been reported before in other BCC metals[39,43]. It is worth noting that the local displacement during the BCC-ω transition is the same as a Burgers vector in the <111>{$\bar{2}$11} system, which suggests that dislocation

mobility and phase stability are correlated. In other words, the electronic structure of HCP elements, which promotes the above phase transformations, also contributes to dislocation mobility (and, in turn, ductility). Within the wide compositional space of single-phase BCC solid solutions, these features are likely to be universal allowing alloy designers to systematically vary the different parameters identified here to optimize strength and ductility. Returning to the equiatomic VNbMoTaW and TiZrNbHfTa alloys, these alloys contain 0 at% and 60 at% of HCP elements. Based on the trends above, it is likely that the unique slip anisotropy (preference for ($\bar{2}11$) slip) observed in TiZrNbHfTa is caused by an essential property of dislocation motion in RHEAs with high HCP-element concentrations.

## Discussion

Two model BCC RHEAs, brittle VNbMoTaW and ductile TiZrNbHfTa, were investigated to better understand fundamental aspects of their mechanical behavior. Bulk and micropillar compression tests as a function of temperature and strain rate show the latter is plastically deformable in compression down to 77 K, but the former is not below room temperature. From our experimentally determined enthalpies of compression, the BDTT of VNbMoTaW is estimated to be 627 K (above room temperature) whereas that of TiZrNbHfTa is estimated to be 247 K (below room temperature) which indicates that high dislocation mobility at room temperature is one reason for the latter's more ductile behavior. Consistent with this, deformation enthalpies and strength models both show kink migration is the rate-controlling step at room temperature in VNbTaMoW but not in TiZrNbHfTa. Another reason for the greater malleability of TiZrNbHfTa is the extremely low energy of screw dislocation cores in this RHEA compared with those in VNbMoTaW. Consequently, it is easy to nucleate and activate many dislocations in TiZrNbHfTa, even at low temperatures. After plastic deformation, screw dislocations dominate the microstructure in TiZrNbHfTa (like in conventional BCC metals), but screw and edge segments are present roughly equally in VNbTaMoW. These results can be explained by results of our strength modeling which show that screw and edge dislocation strengthening are similar in magnitude in VNbMoTaW whereas screw dislocation strengthening is greater in TiZrNbHfTa.

The slip behaviors of the two RHEAs are distinctly different: ($\bar{1}01$) slip is preferred in VNbMoTaW as observed commonly in BCC alloys, whereas ($\bar{2}11$) slip is preferred in TiZrNbHfTa. First-principles calculations of virtual BCC alloys reveal that the persistent slip on ($\bar{2}11$), unique to TiZrNbHfTa, is likely related to changes in electronic structure attributable to alloying with HCP elements in relatively high concentrations. That said, first-principles calculations have size limitations when it comes to addressing aspects of macroscopic slip (plasticity), which are generally more tractable by MD simulations. However, classical MD simulation cannot treat these multi-component alloys due to the lack of reliable interatomic potentials for the quinaries. An alternative is to consider the energy barrier of dislocation motion, which can provide clues to understanding slip behavior. Since direct calculation of the energy barrier for five component RHEAs is not possible, we performed first-principles calculations for binary systems based on virtual crystal approximation (VCA). This is the only way to treat dislocation motion in alloy systems at this stage of first-principles calculations.

However, first-principles calculations could be used to evaluate most of the important structural, elastic, and energetic features (such as lattice constants, elastic constants, local lattice distortion, and dislocation core structure and energy) of the equiatomic VNbTaMoW and TiZrNbHfTa alloys. These calculations involved the actual quinary RHEAs (and not just the binaries). Together with experimental results, these calculations show that the higher absolute yield strength of VNbTaMoW relative to TiZrNbHfTa is due to its stronger dislocation-solute interaction energy, greater vacancy and interstitial formation

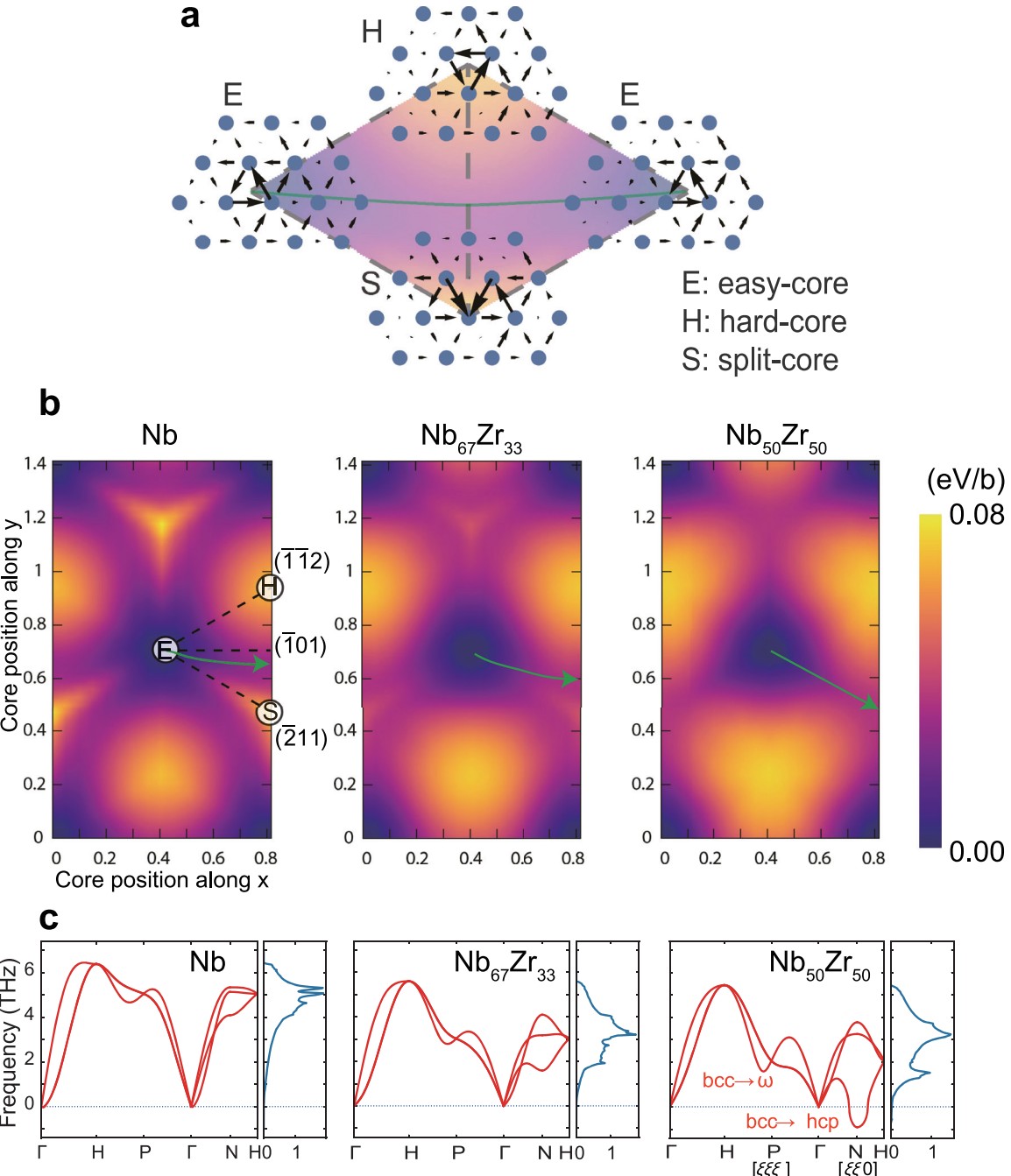

**Fig. 4 | Two-dimensional Peierls potential energy surface for the motion of a screw dislocation. a** Schematic image of dislocation core structures associated with the Peierls potential surface in BCC metals. The symbols E, H, and S correspond to the easy-, hard-, and split-core configurations, respectively. **b** Peierls potential energy surface in pure Nb, virtual $Nb_{67}Zr_{33}$, and virtual $Nb_{50}Zr_{50}$ alloys, where the energy was calculated for several sampling configurations within the E,

H, and S triangles. The arrows correspond to the minimum energy path for the dislocation motion. **c** Phonon dispersion and density of states of pure Nb, virtual $Nb_{67}Zr_{33}$, and virtual $Nb_{50}Zr_{50}$ alloys, where phonons along high-symmetry directions in the Brillouin zone of bcc phase are shown, where the labels Γ, H, P, and N corresponds to high-symmetry points.

energy, and higher shear modulus. First-principles calculations show that the higher concentration of HCP elements and their contribution to the electronic structure of the alloys is responsible for the lower shear modulus of TiZrNbHfTa. When normalized by their respective shear moduli, the yield strength of TiZrNbHfTa is higher than that of VNbMoTaW. This is due to the former's significantly greater lattice distortion, which we attribute to a higher degree of local atomic shuffling induced by the HCP elements. The greater lattice distortion in TiZrNbHfTa is related to its tendency for phase instability, which makes the dislocation core in TiZrNbHfTa more extended than the

compact core in VNbTaMoW. Our findings demonstrate promising ways to control the mechanical properties and broaden the utility of refractory alloys by tuning the concentration of HCP elements.

## Methods
### Experimental procedures
Ingots of TiZrNbHfTa and VNbMoTaW equiatomic alloys were prepared by arc-melting the constituent elements (>99.9% pure) in argon atmosphere. The TiZrNbHfTa alloy was rolled at room temperature to a thickness reduction of 50% and then annealed at 1200 °C for 144 h,

followed by water quenching. As the VNbMoTaW alloy is too brittle to be rolled, it was annealed at ~2000 °C (temperature estimated from the xenon lamp power) for 48 h in an optical floating-zone furnace in flowing Ar. BCC single-phase microstructures were obtained for the TiZrNbHfTa and VNbMoTaW equiatomic alloys with average grain-sizes of 424 and 496 μm, respectively (Supplementary Fig. 1).

Specimens for mechanical tests were cut by spark-machining. The dimensions of the specimens used for compression tests were $2 \times 2 \times 5$ mm³. Prior to compression tests, specimen surfaces were mechanically polished with diamond paste, followed by electro-polishing with a solution of perchloric acid and methanol (1:9 by volume) for the TiZrNbHfTa alloy and by polishing with colloidal silica for the VNbMoTaW alloy. Compression tests were conducted on an Instron-type testing machine in vacuum at temperatures in the range 77–1073 K at an engineering strain rate of $1 \times 10^{-4}$ s⁻¹. Yield stresses were determined by the 0.2% offset method. For some specimens, the strain-rate sensitivity of flow stress was measured by suddenly changing the strain rate by an order of magnitude at selected temperatures. Deformation markings on the specimen surface were examined by light microscopy and scanning electron microscopy (SEM) utilizing electron backscatter diffraction (EBSD).

Micropillar compression tests were conducted as a function of crystal orientation with a flat punch indenter tip using an Agilent Technologies Nano Indenter G200 nanomechanical tester at room temperature in the displacement-rate-controlled mode at a nominal strain rate of $1 \times 10^{-4}$ s⁻¹. Micropillar specimens with a square cross-section having an edge length $L$ ranging from 0.8 to 10 μm and aspect ratios of approximately 1:2–1:3 were fabricated on a colloidal silica polished surface with a JEOL JIB-4000 focused ion beam (FIB) machine at an operating voltage of 30 kV. Prior to FIB machining, crystal orientations of grains on the polished surface were analyzed by EBSD in a JEOL JSM-7001FA SEM equipped with a field-emission gun operated at 20 kV. Micro-structures of micropillar specimens before and after compression tests were examined by SEM with a JEOL JSM-7001FA electron microscope. The ψ-χ relation (Fig. 2) is usually used to deduce the preference of slip plane in BCC metals: {110}, {112}, or MRSS (maximum resolved shear stress) planes[30,44–46]. Slip plane preference is considered to be closely related to the dislocation core structure, and hence is a fundamental property of plastic deformation.

Fine deformation microstructures after bulk and micropillar compression tests were examined by transmission electron microscopy (TEM) and scanning transmission electron microscopy (STEM) with JEOL JEM-2000FX and JEM-2100F electron microscopes operated at 200 kV, respectively. Thin foils for TEM/STEM observations were prepared by electro-polishing in a solution of nitric acid, ethylene glycol, and methanol (2:5:20 by volume). The specimens for TEM/STEM observations were prepared by the FIB-SEM in-situ lift-out technique using a FEI Quanta 3D 200i Dual-Beam system equipped with an Omniprobe nanomanipulator.

The ψ-χ relation (Supplementary Fig. 2a) was investigated at room temperature (both VNbMoTaW and TiZrNbHfTa) and 77 K (only TiZrNbHfTa) using polycrystals deformed in compression at a strain rate of $1 \times 10^{-4}$ s⁻¹ to 1-2% plastic strain. Macroscopic slip planes were determined by trace analysis made on two orthogonal side surfaces with EBSD (Supplementary Figs. 2b, c).

Potential effects of short-range ordering (SRO) and short-range clustering (SRC) were not considered in the present study although their presence is likely (but not confirmed) in our RHEAs. Our reasoning was as follows. A recent study[47] showed that even when SRO was present in a FCC medium-entropy alloy (CrCoNi), it had no measurable effect on the yield strength or slip behavior. We suspect that was due to the high friction stress of CrCoNi in the random state (i.e., in the absence of SRO) making any additional effect of SRO negligible.

Since the friction stress of the RHEAs investigated here is much higher than that of CrCoNi, we expected SRO to have an even smaller effect and therefore did not explicitly look for it.

## Yield strength modeling

Screw dislocation strengthening due to solutes is predicted to approximately follow the proportionality,

$$\sigma_y \propto \left( \sum_i c_i E_i^2 \right)^{2/3} \tag{1}$$

where $c_i$ and $E_i$ are the concentration of the solute $i$ and its interaction energy with the screw dislocation core. The constant of proportionality, even though termed a constant, is dependent on the average vacancy and self-interstitial formation energies of the alloy and increases with increasing values of these energies. Similarly, edge dislocation strengthening is predicted to be approximately proportional to[32,48]

$$\sigma_y \propto \langle \mu \rangle \left( \sum_i c_i \delta V_i^2 \right)^{2/3} \tag{2}$$

where $\delta V_i$ is the misfit volume of the solute $i$ in the average alloy and $\langle \mu \rangle$ is the average shear modulus of the alloy.

Several models based on the motion of either screw or edge dislocations have been developed to describe yield behavior in high-entropy alloys[49–51]. The mechanisms of strengthening considered here are the kink migration barrier on screw dislocations as well as the stress to overcome edge dipoles. Edge dipoles are extended on screw dislocations as they move at jogs formed due to kink-kink collisions between kink-pairs formed on different {110} planes. The key quantities required to evaluate screw strengthening are the various solute-screw dislocation interaction energies in the average alloy as well as the average vacancy and self-interstitial formation energies. The vacancy and self-interstitial formation energies are required to determine the edge dipole barrier, whereas the solute-screw dislocation interaction energies are required to determine the kink migration barrier. The average vacancy and self-interstitial formation energies in the random high-entropy alloy are calculated using a rule of mixtures from their values in the individual element BCC lattices, Supplementary Table 6[32,52]. The computed average vacancy and self-interstitial formation energies are higher in VNbMoTaW (3.03 and 6.28 eV) than in TiZrHfNbTa (2.28 and 3.79 eV). As a result, the proportionality factor in Eq. 1 for VNbMoTaW is greater by 25% than in TiZrHfNbTa which, together with the terms in the brackets, leads to a difference of 442 MPa in their computed yield stresses from screw strengthening. The various solute-screw dislocation interaction energies are evaluated in an approximate fashion using Zhou's embedded atom method (EAM) potentials developed for approximately 21 elements in the periodic table[53–55]. Here, we use such potentials with modified Nb-W, Nb-Mo, Ta-W, and Ta-Mo cross interactions. The various solute-screw dislocation interaction energies for the TiZrNbHfTa system have been reported in an earlier study[48] and are reproduced in Supplementary Table 2. Supplementary Table 3 gives the various interaction energies between solutes and screw dislocations for the VNbMoTaW system evaluated using the EAM potentials in this study. Yield strength models based on edge dislocation motion for these alloys evaluate the activation barrier for edge dislocations to move in a random field of solutes[48].

## Simulation methods

First-principles calculations were carried out in the simulation study to investigate fundamental mechanical properties and dislocation core structures. Atomic models of equiatomic VNbMoTaW and TiZrNbHfTa alloys with BCC structure were considered. To construct a statistically

random solid solution, special quasi-random structures (SQS) were generated using the "mcsqs" function in the alloy theoretic automated toolkit (ATAT)[56]. At first, a 250-atom supercell with dimensions of $5 \times 5 \times 5$ lattice units along [100], [010], and [001] was considered to evaluate fundamental properties such as the lattice constant, elastic constants, and local lattice distortion. Ten different random models were prepared to obtain an average of the above properties.

Subsequently, the core structures of a screw dislocation were explored within the first-principles framework. For that, a dislocation dipole was inserted into a 405-atom supercell, with dimensions $\mathbf{a} = 5\mathbf{e}'_1$, $\mathbf{b} = 2.5\mathbf{e}'_2 + 4.5\mathbf{e}'_2(+0.5\mathbf{e}'_3)$, and $\mathbf{c} = 3\mathbf{e}'_3$, where a coordinate system corresponding to $x = [11\bar{2}]$, $y = [1\bar{1}0]$ and $z = [111]$ directions are defined, and vectors in the system are defined as $\mathbf{e}'_1 = a_0[\sqrt{6}, 0, 0]$, $\mathbf{e}'_2 = a_0[0, \sqrt{2}, 0]$, and $\mathbf{e}'_3 = a_0[0, 0, \sqrt{3}/2]$, with lattice constant $a_0$. It has been shown that fundamental properties of screw dislocations in BCC metals can be reproduced well when a dipole is arranged in the above configuration[57]. As the cumulative displacement component along the $\mathbf{c}$ direction appears according to the elasticity theory of the dislocation dipole configuration in a periodic cell[58], the component of $0.5\mathbf{e}'_3$ is added to the unit cell vector of $\mathbf{b}$. The positions of atoms in the supercell were determined according to the BCC crystal coordinate system. The energy of dipole used for the normalization of core energy of dislocation from first-principles calculations shown in Fig. 3e was evaluated by the elasticity theory.

First-principles calculations were implemented using the Vienna ab initio simulation package (VASP)[59,60]. Projector augmented wave potentials[61] were employed with the Perdew–Burke–Ernzerhof generalized gradient approximation exchange-correlation density functional[62]. The Brillouin-zone gamma-centered k-point samplings were chosen using the Monkhorst–Pack algorithm[63], where Γ-centered $3 \times 3 \times 3$ and $1 \times 1 \times 6$ grids were used for 250-atom and 405-atom models, respectively. A cutoff in plane-wave energy of 400 eV was applied using a first-order Methfessel–Paxton scheme that employed a smearing parameter of 0.1 eV. The total energy was converged for all calculations within $10^{-6}$ eV/atom. The relaxed configurations were obtained using the conjugate gradient method that terminated the search when the force on all atoms was reduced to 0.01 eV/Å. The Virtual Crystal Approximation (VCA) in QUANTUM ESPRESSO package[64] was also employed to evaluate the dislocation core properties of the virtually mixed state of the Nb-Zr alloy system. The optimized parameters for kinetic energy cutoff of 52 Rydberg and Γ centered $1 \times 1 \times 18$ k-point grids were used. Davidson's iterative diagonalization and Broyden–Fletcher–Goldfarb–Shanno algorithm were applied for the self-consistent field of electronic structure and structural relaxation, respectively. The phonon dispersions of virtual alloys were evaluated using phonopy software[65]. The atomic configurations were visualized using VESTA and AtomEye software[66,67].

## Data availability

The authors declare that all data supporting the findings of this study are available within the article and its Supplementary Information. All other data are available from the corresponding author upon request.

## Code availability

No custom software was used during the current study. The scripts for dislocation core analysis are available from the corresponding authors upon request.

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

## Acknowledgements

This work was supported by Grant-in-Aids for Scientific Research on Innovative Areas on High-Entropy Alloys through the grant numbers JP18H05450 (H.I.), JP18H05451 (H.I.) and JP18H05453 (T.T.), in part by JSPS KAKENHI (grant numbers JP20K21084 (K.K.), JP21H01651 (K.K.), JP22H00262 (H.I.), JP22H01762 (T.T.) and JP23KJ1302 (S.H.)), and JST CREST, PRESTO and FOREST (grant numbers JPMJCR1994 (K.K.), JPMJPR1998 (T.T.) and JPMJFR213P (T.T.)). T.T acknowledges Prof. D.C.

Chrzan for a fruitful discussion on phase stability of BCC-HCP transition. Simulations were performed on the large-scale parallel computer system of HPE SGI 8600 at JAEA.

## Author contributions

H.I. conceived and directed the project. S.H., S.M., Z.C., and K.K. conducted bulk and micropillar compression tests and SEM and TEM observations. S.I.R. performed the yield strength modeling. T.T. and I.L. performed the first-principles calculations. T.T., K.K., S.I.R., C.W., E.P.G., and H.I. analyzed the experimental and theoretical data to develop the conceptual underpinnings of this paper. T.T., S.H., S.I.R., C.W., E.P.G., and H.I. drafted the manuscript. All authors discussed the results and commented on the manuscript. T.T., K.K., H.I., and E.P.G. contributed equally to this work.

## Competing interests

The authors declare no competing interests.
