## [Peer Review File · Nature Communications]

Intrinsic factors responsible for brittle versus ductile nature of refractory high-entropy alloysREVIEWER COMMENTS

Reviewer #1 (Remarks to the Author):

This paper offers a comprehensive discussion on the strength and ductility of two representative high entropy alloys (VNbMoTaW and TiZrNbHfTa) using a blend of mechanical tests at different temperatures, micropillar tests, post-mortem dislocation analysis, dislocation core energy calculation, and electronic structure calculation. It is noteworthy that every experiment in this paper is systematic and of very high quality, particularly given the difficulty in creating a fully homogenized VNbMoTaW sample, which forms the foundation for the subsequent in-depth analysis. The discussions, derived from both experimental and simulation data, are detailed and well-articulated, integrating cutting-edge theoretical works and their own innovative findings. The work astutely explores numerous factors influencing the behavior of screw and edge dislocations, and aptly elucidates why TiZrNbHfTa possesses higher ductility despite high normalized yield strength by understanding the importance of (-211) slip in TiZrNbHfTa. Although refractory HEAs have been extensively studied, there still exists a gap in the community for a comprehensive investigation of these alloys' strength and ductility from experimental to theoretical perspectives. As such, I concur that this work is highly impactful and aptly suitable for a prestigious publication like Nature Communications. Nonetheless, I suggest some revisions to further refine the manuscript before recommending its acceptance.

- (1) The authors derived the BDTT from the strain rate jump tests. Can they correlate this with macroscopic results, such as the results from compression tests? Could they also shed some light on the high-temperature behavior deviating from the $H=35kT$ plot?
- (2) For reader clarity, it would be beneficial to annotate screw and edge dislocations in Figs. 2 ijkl, as demonstrated in Lu et al., PNAS, 118 (2021).
- (3) The authors employed different models for calculating yield strengths. However, the manuscript primarily mentions proportional relationships, making it challenging to quantitatively comprehend the authors' arguments. For instance, the squared summation of the solute-screw dislocation interaction energies of TiZrNbHfTa using equation (1) is only 20% smaller than that of VNbMoTaW, while the authors' cited difference is 442 MPa. Consequently, it is unclear what other factors contribute to this large difference besides the interaction energy (possibly melting temperature or shear modulus?). Therefore, providing complete equations, tables, and more comprehensive explanations regarding the calculation would be beneficial.
- (4) The authors linked the magnitudes of screw and edge dislocation strengthening to the post-deformation dislocation structures of VNbMoTaW. So, what are the implications of these magnitudes on the screw dislocation-dominant deformation of TiZrNbHfTa? The difference is only 102 MPa, making it tough to comprehend the screw dislocation-dominant deformation of TiZrNbHfTa from the strengthening model. Can the authors comment on this?
- (5) The authors utilized DFT-calculated shear moduli for calculating the modulus-normalized yield stress of both alloys. More detailed rationale for using G along $\langle 111 \rangle$ on $\{110\}$ plane would be appreciated, rather than relying on experimental values (e.g., TiZrNbHfTa: 35.8 GPa (Laplanche et al., Journal of Alloys and Compounds, 799 (2019)) or other shear moduli, such as C44, commonly used for other alloys. This clarification won't change the discussion but will assist readers in understanding the authors' considerations, especially since the authors consider the crystallographic orientation effects.
- (6) Can the authors share their thoughts on the implications of the distributed core energy of TiZrNbHfTa? Will these variations affect any specific properties?
- (7) The role of valence electron concentrations in affecting ductility has been the focus of prior studies, for example, in the context of Jahn-Teller distortion (Qi et al., Physical Review Letters 112 2014). Thus, elements with low VEC have also been used in the development of new BCC HEAs. Despite the difference in fundamental reasoning, the direction of alloy design appears to converge. Could the authors elucidate any fresh perspectives on alloy design that their novel proposal brings to light?
- (8) While the impact of incorporating HCP elements on ductility is compelling, the effect on high shear-normalized yield strength, as mentioned in line 41, is not as clear, especially given that it is dominated

by screw dislocation. Could the authors clarify whether the inclusion of HCP elements influences the solute-screw interaction energy and, if so, in what manner does it modify this energy?

(9) For consistency, it might be better to change DBTT to BDTT in line 254.

(10) At line 344, the authors posit that the electronic structure of the alloys contributes to the lower shear modulus of TiZrNbHfTa. I suspect this references the local atomic shuffling deliberated on page 12. However, it would be advantageous for the authors to explicitly clarify this point.

(11) Given the certainty of their presence in these alloys, it might be beneficial for the authors to discuss the potential influence of short-range ordering or clustering on these alloys in light of their theoretical framework.

Reviewer #2 (Remarks to the Author):

The manuscript "Why some refractory high-entropy alloys are brittle and others ductile" is well written and tackles a very intriguing and relevant problem. A good combination of experiments and simulation was performed, and the results are relevant. Although I feel the manuscript sheds light on the issue, I do not feel it answers the question presented on the title. Therefore, I feel the manuscript should be considerably reworked for it to be published.

The main key point that I feel needs better explanation is exactly regarding the question being asked in the title. The authors show and justify the brittleness of RHEAs based on the dislocation behavior, activated slip planes and dislocation core structure and type. The conclusion is on the same direction as some previous publications that concluded that using "lower VEC" elements such as Ti, Zr and Hf would lead to a more ductile behavior, however the explanation presented in this manuscript is much more fundamental. However, there are still two main problems in my opinion:

1 – The title presents a very generic question, and the answer still seems very specific in my opinion. The authors in the end only suggest that this behavior will translate into all other RHEAs, local interactions or other unforeseen effects might arise for other compositions. Therefore, for the authors to publish this paper on a high impact journal such as Nature, I feel they need to somehow present a trend, showing that multiple other alloys from the literature should follow this trend, and how their new theory can explain the trend for ductility in other alloys as well.

2 – One key point for fracture is that it is not only dictated by the mobility of dislocations, instead, it is often more critically controlled by defects, especially brittle fracture. This is one key point that in my opinion many publications on this topic are missing. This is very relevant in the context of this publication because basically all the explained phenomena should impact dislocation mobility, and thus strength but what about the fracture as a whole? I assume the authors assume that all samples do not have any "macroscopic defect", since they are all micropillars, so what makes an alloy intrinsically more brittle? Is it only being stronger? Is there anything other than that? Because by this optic, to make an alloy ductile, we only need to make it weaker. However, without doing a deep literature survey, it would be hard to make this claim. If there is something other than strength alone, this would be very desirable to be understood. Are there alloys that can be intrinsically more ductile, so they can resist higher strengths/defects and others can't?

Other minor points that I suggest to be addressed:

Page 5 – please explain better the statement in lines 125-127.

Page 10 – lines 238, 239 – dislocation dipole is a somewhat generic terminology. Please describe in detail which type of dipole was built and also why specifically a dislocation dipole was chosen for this investigation;

also in this paragraph – you mention that the lower nucleation energy for the loops explain the better ductility, going back to my main point number 2, would you claim this only by the strength argument or do you hypothesize that it could have any further impact?

Figure 3 is a bit confusing to me, you are inverting the order, c relates to f and d to e, correct? It

would be better to put each result in a separate column. Also please specify the temperature in the caption (0 Kelvin?)

Reviewer #3 (Remarks to the Author):

Tsuru et al provided an interesting and detailed mechanical property study of TiZrHfNbTa and VNbMoTaW MPEAs, and binary systems to explain some underlying mechanisms controlling mechanical behavior. The manuscript is well written and easy to read. I appreciate the good amount of effort authors put in this work.

The title of this manuscript "Why some refractory high-entropy alloys are brittle and others ductile" made me very curious about knowing more about ductility in BCC refractors, which is very critical in any alloy for practical application. But the overall message changed to other mechanical properties over ductility. I really don't see that manuscript is justifying the title.

Major concerns:

(1) In Fig. 1, author didn't outline if its compressive or tensile yield stress? This is important because all the experimental tests are compressive.

(2) In general, minimum of five independent slip systems are required for inducing ductility in a polycrystalline material.

Although, BCC systems have too many slip systems(48) that interfere with each other and mutually obstruct slip movement, which makes it difficult to have tensile elongation. Thus, we don't expect much ductility in hcp, i.e., they are also brittle. Similarly, HCP has three slip systems (less than five slip systems), so we hardly see any ductility in HCP systems too.

In the present manuscript, the mechanism of mixed bcc+hcp giving rise to higher ductility is still not clear to me.

(3) What is the compressive strain for these two alloys? It is not mentioned anywhere except in figure 2. The results are only for 2-3% strain. How to quantify the increase in ductility in VNbMoTaW vs TiZrNbHfTa?

(4) I'm not sure I have a good way of comparing different loading direction results in Fig. 2. Author's need to explain it with reason.

(5) This work provides an electronic structure changes related to the ratio of HCP to BCC to control mechanical properties. Given the manuscript is about ductility, I don't see discussion or quantification of ductility in the manuscript, in terms of quantification by providing numbers?

(6) Author's suggest for the next generation of high-temperature materials, however, I don't think without good tensile ductility such alloy can be useful for good practical application.

(7) To me, the bright-field TEM are not good enough to understand dislocation structure. Needs better TEM analysis.

(8) Binary vs MPEAs are completely different in behavior, is it that straight forward to translate Fig. 4 results to MPEAs, this is highly over simplification.

For the reasons listed as above, I do not think the quality of this manuscript reaches the level of

Nature Commun. However, overall the manuscript is well written, and should be published in more specialized journal.

July 19, 2023

Response To Reviewers for Nature Communications

Ref. No: NCOMMS-23-14936-T

We thank the reviewers for the valuable feedback on our manuscript.

Following your recommendation, we have performed additional work, both experimental and theoretical, to strengthen the study. As a result, we can now provide quantitative arguments for the trends seen in the two model alloys of this study and discuss the generality of these trends. We have also changed the title to: “Reasons for the differences in mechanical properties of two model refractory high-entropy alloys: brittle VNbMoTaW and ductile TiZrNbHfTa” to better reflect the content of the paper.

In what follows, we provide a point-by-point response to each comment (the corresponding changes made to the manuscript are highlighted in blue).

Reviewer #1 (Remarks to the Author):

This paper offers a comprehensive discussion on the strength and ductility of two representative high entropy alloys (VNbMoTaW and TiZrNbHfTa) using a blend of mechanical tests at different temperatures, micropillar tests, post-mortem dislocation analysis, dislocation core energy calculation, and electronic structure calculation. It is noteworthy that every experiment in this paper is systematic and of very high quality, particularly given the difficulty in creating a fully homogenized VNbMoTaW sample, which forms the foundation for the subsequent in-depth analysis. The discussions, derived from both experimental and simulation data, are detailed and well-articulated, integrating cutting-edge theoretical works and their own innovative findings. The work astutely explores numerous factors influencing the behavior of screw and edge dislocations, and aptly elucidates why TiZrNbHfTa possesses higher ductility despite high normalized yield strength by understanding the importance of (-211) slip in TiZrNbHfTa. Although refractory HEAs have been extensively studied, there still exists a gap in the community for a comprehensive investigation of these alloys' strength and ductility from experimental to theoretical perspectives. As such, I concur that this work is highly impactful and aptly suitable for a prestigious publication like Nature Communications.

We thank the reviewer for the positive comments about our work and its impact.

Nonetheless, I suggest some revisions to further refine the manuscript before recommending its acceptance.

Comment (1): The authors derived the BDTT from the strain rate jump tests. Can they correlate this with macroscopic results, such as the results from compression tests? Could they also shed some light on the high-temperature behavior deviating from the $H=35kT$ plot?

Response to reviewer #1's comment (1):

To answer the reviewer's question, we performed additional compression tests on VNbMoTaW and TiZrNbHfTa. Stress-strain curves from those tests are shown below, and in Fig 1 (a, b) of the revised manuscript (details of the tests are provided in the Methods section). At room temperature, TiZrNbHfTa exhibits extensive deformability, consistent with previous results [3]. In contrast, VNbMoTaW exhibits limited plastic deformability at room temperature, again consistent with previous results [4]. Taken together, these results are consistent with the BDTTs deduced from the strain rate jump tests: 247 K for TiZrNbHfTa and 627 K for VNbMoTaW, i.e., a few tens of degrees below room temperature and a few hundreds of degrees above room temperature, respectively. Importantly, when compressed at a temperature higher than its BDTT, VNbMoTaW exhibits good deformability. Thus, the answer to the reviewer's first question is yes, the BDTTs estimated from the strain rate jump tests correlate well with those from macroscopic mechanical tests. As for the reviewer's second question regarding what might be going on at higher temperatures where there is a deviation from the $H=35kT$ line, all we can say is that kink-pair migration likely ceases to be the rate-limiting mechanism and others such as dislocation multiplication and motion become important (see our answer to reviewer #2). Additional work (beyond the scope of the present paper) is needed to truly pin down those governing mechanisms; hence we are hesitant to speculate at this stage.

Fig. 1 | Compressive mechanical properties of equiatomic VNbMoTaW and TiZrNbHfTa RHEAs. a,b, Stress-strain curves for VNbMoTaW (a) and TiZrNbHfTa (b) compressively deformed at temperatures below and above their respective BDTTs. Fracture occurred in all cases, except in TiZrNbHfTa at room temperature; the corresponding fracture strains (ϵ_f) are labeled in the plots.

Line 96:

Examples of compressive stress-strain curves at temperatures below and above the respective brittle-to-ductile transition temperatures (BDTTs) are shown in Figs. 1a,b. Consistent with earlier reports [3,16,17], VNbMoTaW is brittle while TiZrNbHfTa is ductile at room temperature.

As additional information, the extended stress-strain curve of TiZrNbHfTa at room temperature is shown below. The sharp rise in stress at high strains occurs when the aspect ratio of the specimen becomes smaller than 1 at which point the compression platens constrain and block 45° slip. Fracture did not occur in this specimen, and the test was eventually stopped. Only a portion of this stress-strain curve is shown in Fig. 1b, namely, before the platens constrain

the specimen and make the measured stress invalid.

Fig. A Stress-strain curve for TiZrNbHfTa compressively deformed at room temperature.

Comment (2): For reader clarity, it would be beneficial to annotate screw and edge dislocations in Figs. 2 i-kl, as demonstrated in Lu et al., PNAS, 118 (2021).

Response to reviewer #1's comment (2):

We agree with the reviewer that annotation would be helpful to the readers. Accordingly, we have modified Figure 2 and marked some of the screw and edge segments in the TEM images (see below and in revised manuscript).

Fig. 2

Comment (3): The authors employed different models for calculating yield strengths. However, the manuscript primarily mentions proportional relationships, making it challenging to quantitatively comprehend the authors' arguments. For instance, the squared summation of the solute-screw dislocation interaction energies of TiZrNbHfTa using equation (1) is only 20% smaller than that of VNbMoTaW, while the authors' cited difference is 442 MPa. Consequently, it is unclear what other factors contribute to this large difference besides the interaction energy (possibly melting temperature or shear modulus?). Therefore, providing complete equations, tables, and more comprehensive explanations regarding the calculation would be beneficial.

Response to reviewer #1's comment (3):

The reviewer is right that the squared summation of the solute-screw dislocation interaction energies do not fully account for the computed screw strength difference between HfNbTaTiZr and VNbMoTaW. Therefore, the proportionality factor in equation (1) of the Methods section needs to be quantified to get a full measure of screw strengthening in the two alloys. Further details of how they can be computed are given in the references cited in that section. Briefly, the proportionality factor in equation (1) is dependent on the average vacancy and self-interstitial formation energies and increases with increasing values of these energies. The average vacancy and self-interstitial formation energies are higher in VNbMoTaW (3.03 and 6.28 eV) than in TiZrHfNbTa (2.28 and 3.79 eV). These average values were computed using rule of mixtures from the values for the constituent bcc elements (which are given in the table below). Hence, the proportionality factor in equation 1 for VNbMoTaW is greater by 25% than in TiZrHfNbTa which, together with the terms in the brackets, leads to a difference of 442 MPa in their computed yield stresses from screw strengthening.

The following modifications were made to the revised manuscript.

Line 573:

The average vacancy and self-interstitial formation energies in the random high-entropy alloy are calculated using a rule of mixtures from their values in the individual element BCC lattices, Extended Data Table 4 [33,52]. The computed average vacancy and self-interstitial formation energies are higher in VNbMoTaW (3.03 and 6.28 eV) than in TiZrHfNbTa (2.28 and 3.79 eV). As a result, the proportionality factor in equation 1 for VNbMoTaW is greater by 25% than in TiZrHfNbTa which, together with the terms in the brackets, leads to a difference of 442 MPa in their computed yield stresses from screw strengthening.

We also added the following table as Extended Data Table 4.

Extended Data Table 4 | Vacancy and self-interstitial formation energies of the relevant BCC elements [33,52].

Element	Vacancy formation energy (eV)	Self-int formation energy (eV)
Nb	2.99	5.25
Ta	3.14	5.83
Mo	2.96	7.42
W	3.56	9.55
V	2.51	3.31
Ti	1.55	2.33

Hf	2.0	3.0
Zr	1.7	2.55

Comment (4): The authors linked the magnitudes of screw and edge dislocation strengthening to the post-deformation dislocation structures of VNbMoTaW. So, what are the implications of these magnitudes on the screw dislocation-dominant deformation of TiZrNbHfTa? The difference is only 102 MPa, making it tough to comprehend the screw dislocation-dominant deformation of TiZrNbHfTa from the strengthening model. Can the authors comment on this?

Response to reviewer #1's comment (4):

Our rationale for linking the magnitudes of screw and edge dislocation strengthening to the post-deformation dislocation structures was as follows. The calculated edge strengthening to screw strengthening ratio is ~0.87 (less than 1) in TiZrHfNbTa and ~0.98 (close to 1) in VNbMoTaW. We believe, therefore, that more screw segments should be present in TiZrHfNbTa (as they are harder to move) but roughly equal numbers of edge and screw segments should be present in VNbMoTaW. This is indeed what we observe in our TEM foils. Admittedly, these are relative strengths, but we are not aware of a way to determine absolute values for the above ratios at which screws, edges, or both will dominate. In other words, we cannot assert that when the ratio is below a certain value, screws will dominate, and when it is above a certain value, edges will dominate (clearly, though, when the ratio is very close to 1, edges and screws should be equally prevalent). Our computations only yield trends that are consistent with experimental observations.

Comment (5): The authors utilized DFT-calculated shear moduli for calculating the modulus-normalized yield stress of both alloys. More detailed rationale for using G along $[111]$ on $\{110\}$ plane would be appreciated, rather than relying on experimental values (e.g., TiZrNbHfTa: 35.8 GPa (Laplanche et al., Journal of Alloys and Compounds, 799 (2019)) or other shear moduli, such as C_{44} , commonly used for other alloys. This clarification won't change the discussion but will assist readers in understanding the authors' considerations, especially since the authors consider the crystallographic orientation effects.

Response to reviewer #1's comment (5):

Thank you for the opportunity to clarify. The C_{44} component from DFT calculations, 37.4 GPa for TiZrNbHfTa (Fig. 3), compares well with the experimental shear modulus obtained by Laplanche et al. for polycrystalline TiZrNbHfTa (35.8 GPa). Therefore, we agree with the reviewer that our use of DFT-calculated single-crystal values does not change the discussion. (Similar comparisons could not be made for VNbMoTaW as experimental values for this alloy have not been reported to our knowledge.) Regarding the reviewer's broader point about why the G along $\langle 111 \rangle$ on $\{110\}$ obtained by DFT calculations was used to normalize the yield stress, our rationale was as follows. Slip of BCC alloys occurs in the $\langle 111 \rangle$ direction of the $\{110\}$ plane. Furthermore, the macroscopic yield stress σ_{th} is expressed in terms of the orthogonal transformation of the critical resolved shear stress (CRSS) taking into account the grain

orientation; that is, there is a linear relationship between yield stress and CRSS. Therefore, to fairly normalize the yield stress associated with the dislocation-mediated process, the shear modulus along the same orientation as the CRSS should be used, namely G along $\langle 111 \rangle$ on $\{110\}$. Note that this value is the average of the values obtained from ten different random solid solution models. The following changes have been made to the manuscript.

Line 114:

Given that the macroscopic yield stress is expressed by the orthogonal transformation of the critical resolved shear stress (CRSS), i.e., $\sigma_y = G \gamma$, yield stresses can be normalized by the 0 K shear modulus (G), and test temperatures (T) by the melting temperature (T_m), and those results are shown in Fig. 1d. In this study, G along $\langle 111 \rangle$ on $\{110\}$ plane (which is the orientation used to determine the CRSS) was determined at 0 K using first-principles calculations: 74.0 and 28.8 GPa for VNbMoTaW and TiZrNbHfTa (more on this later).

Line 600:

At first, a 250-atom supercell with dimensions of $5 \times 5 \times 5$ lattice units along $[100]$, $[010]$, and $[001]$ was considered to evaluate fundamental properties such as the lattice constant, elastic constants, and local lattice distortion. Ten different random models were prepared to obtain an average of the above properties.

Comment (6): Can the authors share their thoughts on the implications of the distributed core energy of TiZrNbHfTa? Will these variations affect any specific properties?

Response to reviewer #1's comment (6):

We thank the reviewer for raising certain important issues regarding TiZrNbHfTa. First, Fig. 3b shows that the mean square atomic displacement (MSAD) associated with the local lattice distortion in TiZrNbHfTa is much higher than that of VNbMoTaW (as well as other BCC alloys). This should, in turn, make the dislocation core structure also highly distorted, as seen in Fig. 3d. In addition, we found that the HCP elements tend to make the dislocation core dissociate on the (112) plane (as shown in Fig. 4b), which indicates that the dislocation core structure is affected by how HCP elements are distributed around the core. In other words, the local configuration of elements significantly changes the dislocation core structure. This is likely why the core energies display a wide distribution (spread).

Another important aspect that we believe is an outcome of the above features of TiZrNbHfTa is that the energy difference between the dislocation core structure and the bulk (undislocated) region becomes smaller, as confirmed in Fig. 3e. This is because the energy of the bulk region in TiZrNbHfTa is already relatively high because of the significant local distortion there. Unfortunately, due to the limitations of our computational resources, we have not been able to obtain statistically valid energy barriers for dislocation motion. However, we hypothesize that the shallow valleys in the energy landscape through which dislocations move contribute to “smoother” plastic deformation (easier glide), thereby contributing to higher ductility. We are currently developing a machine learning potential to evaluate the energy barrier of dislocation motion. In the future, we will use those new simulations to assess this hypothesis. Because these latter aspects are still in the early stages of development, we have chosen not to refer to them in the current paper.

Comment (7): The role of valence electron concentrations in affecting ductility has been the focus of prior studies, for example, in the context of Jahn-Teller distortion (Qi et al., Physical Review Letters 112 2014). Thus, elements with low VEC have also been used in the development of new BCC HEAs. Despite the difference in fundamental reasoning, the direction of alloy design appears to converge. Could the authors elucidate any fresh perspectives on alloy design that their novel proposal brings to light?

Response to reviewer #1's comment (7):

We thank the reviewer for these helpful remarks and directing us to a relevant reference, which we have now included in the revised manuscript. As the reviewer points out, adding alloying elements that lower the VEC has been shown to enhance the intrinsic ductility of BCC metals such as Mo [40]. In that case, the underlying mechanism was postulated to be symmetry breaking by a Jahn-Teller distortion leading to shear instability (dislocation nucleation at the theoretical shear strength) occurring before cleavage (at the theoretical tensile strength). Coincidentally, a couple of the elements that lowered the VEC in that study were also HCP (Ti and Zr). In that sense, there is similarity with our present findings in TiZrHfNbTa where the same HCP elements contribute to ductility. Our view of the contribution of the electronic structure of HCP elements overlaps to some extent those expressed in the above studies. However, our work offers the following additional perspectives (discussed in more detail in the manuscript). The HCP elements contribute to greater lattice distortion both near the dislocation core and in the lattice far away. They also make the dislocation core less compact and decrease the core energy making dislocation nucleation easier. Together, this opens the possibility of designing for both strength and ductility simultaneously. Our work also offers hints that the valleys in the energy landscape might become shallower with the addition of HCP elements, making dislocation motion correspondingly easier. Viewed together, the picture becomes rather complex, and we are unable at present to conjecture whether they are all relatable ultimately to a single parameter such as the VEC. It is intriguing, though, that symmetry breaking via a tetragonal distortion (and phonon softening which is often a precursor to phase instability) in the previous study [40], and a BCC-HCP phase instability in the present study, seem to be common threads underlying the mechanisms driving improved ductility. This aspect is worth further investigation in follow-on studies.

The following sentences were added to the revised manuscript:

Line 387:

A theoretical ductility criterion in BCC alloys has been proposed using a virtual crystal approximation (VCA) scheme, where affine deformation was applied to evaluate the relationship between mechanical response and electronic structure [40]. In the present study, virtual Nb-Zr alloys were constructed with different compositions: pure Nb, Nb67Zr33, and Nb50Zr50, to test our hypothesis regarding the effect of increased HCP-element concentration. Dislocation trajectories were investigated for these compositions by first-principles calculations with the VCA scheme (see Methods).

40. Qi, L. & Chrzan, D. C. Tuning Ideal Tensile Strengths and Intrinsic Ductility of bcc Refractory Alloys,

Comment (8): While the impact of incorporating HCP elements on ductility is compelling, the effect on high shear-normalized yield strength, as mentioned in line 41, is not as clear, especially given that it is dominated by screw dislocation. Could the authors clarify whether the inclusion of HCP elements influences the solute-screw interaction energy and, if so, in what manner does it modify this energy?

Response to reviewer #1's comment (8):

We thank the reviewer for raising an important point. The high shear-normalized yield strength of TiZrHfNbTa arises from the HCP elements decreasing its shear modulus and increasing its local lattice distortion. However, we agree with the reviewer that, from a classical standpoint, it is difficult to comprehend the effect of the volumetric strain component on strengthening given that screw dislocations dominate plastic deformation (more on this later). Meanwhile, we believe that components other than the volumetric strain also play a role in strengthening due to the local lattice strain. To clarify this, we have attempted to visualize the lattice strain using the Green-Lagrange strain tensor.

First, we define the deformation gradient \mathbf{F} using the local displacement of atoms in the HEA relative to the ideal BCC lattice points (in defect-free material), where the local atomic displacement can be evaluated by first-principles calculations.

The Cauchy-Green deformation tensor \mathbf{C} is denoted by $\mathbf{C} = \mathbf{F}^T \mathbf{F}$, and the Green-Lagrange strain tensor \mathbf{E} by $\mathbf{E} = \frac{1}{2}(\mathbf{C} - \mathbf{I})$. With these, it is possible to visualize the local lattice strain introduced by

atomic displacements from the ideal lattice positions. The shear components of the Green-Lagrange strain tensor \mathbf{E} for VNbMoTaW and TiZrNbHfTa are shown below. As can be seen, the lattice distortion due to the HCP elements also influences the broad distribution of shear strain in TiZrNbHfTa.

We believe these shear strain components increase the energy barrier for screw dislocation motion. In other words, the direct effect on the frictional stress of dislocations leads to an increase in strength. In summary, the shear modulus, which is an average (global) property of the system, is decreased by HCP elements. At the same time, the resistance to dislocation motion is increased by the local lattice distortion induced by HCP elements. Together, they result in high modulus-normalized yield strength in TiZrNbHfTa.

We added Extended Data Fig. 5 and the following discussion in the manuscript:

Extended Data Fig. 5 | Local atomic strain based on Green-Lagrange strain tensor. The shear strain components, E_{yz} , E_{zx} , and E_{xy} are visualized for VNbMoTaW and TiZrNbHfTa. These shear strain components contribute to the increase in the energy barrier for screw dislocation motion resulting in high modulus-normalized yield strength in TiZrNbHfTa.

Line 247:

This substantial lattice distortion (large MSAD) contributes to not only the volumetric strain, but also the shear strain components as shown in Extended Data Fig. 5, resulting in the higher modulus-normalized yield stress of TiZrNbHfTa in Fig. 1d.

In addition to the above, our recent calculations [R1] show that the solute-screw dislocation interaction energy has two components: a) Inter-row potential differences between a solute row and the average lattice row [R2], and b) effects due to screw dislocation core formation volume [57]. The latter contribution is a classical PdV term, where P is the pressure field at the screw dislocation core and dV is the volume misfit due to the solute, whereas the former is related to bonding differences between solute and the average lattice rows.

[R1] Rao, S.I. et. al. to be submitted to *Acta Mater.*

[R2] Gilbert, M. R. & Dudarev, S. L. Ab initio multi-string Frenkel-Kontorova model for a $b=a/2[111]$ screw dislocation in bcc Iron, *Philos. Mag.* 90, 1035 (2010).

Comment (9): For consistency, it might be better to change DBTT to BDTT in line 254.

Response to reviewer #1's comment (9):

We agree and have changed DBTT to BDTT for consistency.

Line 268:

This low nucleation energy of dislocation loops contributes to better low-temperature ductility and lower BDTT in TiZrNbHfTa.

Comment (10): At line 344, the authors posit that the electronic structure of the alloys contributes to the lower shear modulus of TiZrNbHfTa. I suspect this references the local atomic shuffling deliberated on page 12. However, it would be advantageous for the authors to explicitly clarify this point.

Response to reviewer #1's comment (10):

Thank you for this helpful comment, which prods us to provide explicit clarification. Basically, our view is twofold: (1) the electronic structure of HCP elements is the main contributor to the lower shear modulus of TiZrNbHfTa, similar to the effect previously proposed for BCC β -Ti alloys [43], and (2) the local lattice distortion is caused by the local displacement components induced by the HCP elements that favor the α - and ω -transformations. As shown in Fig. 4c, the transformations into α and ω phases are promoted by the HCP elements, resulting in a large displacement that cannot be accounted for simply by the difference in atomic radius. In other words, local atomic shuffling leads to large lattice distortion and electronic structure to low shear modulus.

Following the reviewer's suggestion, we have made the following clarification:

Line 307:

This local shuffling leads to the large lattice distortion in TiZrNbHfTa, while the electronic structure of HCP elements contributes to its low shear modulus; the latter effect has been reported before in other BCC metals [40,43].

43. Chrzan, D. C., Sherburne, M. P., Hanlumuang, Y., Li, T. & Morris, Jr., J. W. Spreading of dislocation cores in elastically anisotropic body-centered-cubic materials: The case of gum metal, *Phys. Rev. B* 82, 184202 (2010).

Comment (11): Given the certainty of their presence in these alloys, it might be beneficial for the authors to discuss the potential influence of short-range ordering or clustering on these alloys in light of their theoretical framework.

Response to reviewer #1's comment (11):

Short-range ordering and clustering have been observed in FCC HEAs. As the reviewer notes, and as has been studied using atomistic simulations [R3], SRO can occur also in RHEAs. That said, the influence of SRO on the strength and slip behavior of a FCC medium-entropy alloy (CrCoNi) was recently shown to be so small as to be undetectable in macroscopic tensile tests [Li et al. *Acta Mater.*, 243, 118537, 2023]. We suspect this is due to the high intrinsic friction stress of CrCoNi that overshadows any additional strengthening from SRO. In the BCC HEAs investigated here, the intrinsic friction stress is higher still, which will make the effect of SRO on strength

even more difficult to detect experimentally. By the same reasoning, any slip localization and strain softening resulting from SRO destruction by the passage of dislocations is also likely to be very small, making any potential effect on ductility similarly hard to detect. For these reasons, we did not consider effects of SRO in the present study.

[R3] Antillon, E., Woodward, C., Rao, S. I., & Akdim, B. (2021). Chemical short range order strengthening in BCC complex concentrated alloys. *Acta Materialia*, 215, 117012.

We have added the following to the manuscript.

Line 541:

Potential effects of short-range ordering (SRO) and short-range clustering (SRC) were not considered in the present study although their presence is likely (but not confirmed) in our RHEAs. Our reasoning was as follows. A recent study [47] showed that even when SRO was present in a FCC medium-entropy alloy (CrCoNi), it had no measurable effect on the yield strength or slip behavior. We suspect that was due to the high friction stress of CrCoNi in the random state (i.e., in the absence of SRO) making any additional effect of SRO negligible. Since the friction stress of the RHEAs investigated here is much higher than that of CrCoNi, we expected SRO to have an even smaller effect and therefore did not explicitly look for it.

47. Li, L., Chen, Z., Kuroiwa, S., Ito, M., Yuge, K., Kishida, K., Tanimoto, H., Yu, Y., Inui, H. and George, E.P., *Acta Mater.* 243, 118537 (2023).

Reviewer #2 (Remarks to the Author):

The manuscript “Why some refractory high-entropy alloys are brittle and others ductile”; is well written and tackles a very intriguing and relevant problem. A good combination of experiments and simulation was performed, and the results are relevant. Although I feel the manuscript sheds light on the issue, I do not feel it answers the question presented on the title. Therefore, I feel the manuscript should be considerably reworked for it to be published.

The main key point that I feel needs better explanation is exactly regarding the question being asked in the title. The authors show and justify the brittleness of RHEAs based on the dislocation behavior, activated slip planes and dislocation core structure and type. The conclusion is on the same direction as some previous publications that concluded that using lower VEC elements such as Ti, Zr and Hf would lead to a more ductile behavior, however the explanation presented in this manuscript is much more fundamental. However, there are still two main problems in my opinion:

Comment (1): The title presents a very generic question, and the answer still seems very specific in my opinion. The authors in the end only suggest that this behavior will translate into all other RHEAs, local interactions or other unforeseen effects might arise for other compositions. Therefore, for the authors to publish this paper on a high impact journal such as Nature, I feel they need to somehow present a trend, showing that multiple other alloys from the

literature should follow this trend, and how their new theory can explain the trend for ductility in other alloys as well.

Response to reviewer #2's comment (1):

Upon reflection, we agree that our title was overly broad for this paper. Accordingly, we have changed the title to “Reasons for the differences in mechanical properties of two model refractory high-entropy alloys: brittle VNbMoTaW and ductile TiZrNbHfTa”. We also thank the reviewer for the comment above acknowledging our much more fundamental contributions to the understanding of ductile behavior than previous work. To quote: “*The conclusion is on the same direction as some previous publications that concluded that using lower VEC elements such as Ti, Zr and Hf would lead to a more ductile behavior, however the explanation presented in this manuscript is much more fundamental.*”

We specifically chose two model alloys (one brittle, the other ductile) for investigation. Our goal was to uncover fundamental mechanisms that might explain why these specific alloys behave so differently from each other but also whether more general lessons can be learned that might be applicable (in a predictive way) to alloy design in the future. We believe our efforts were largely successful, as the following summary of our principal findings demonstrate. A decrease in the shear modulus occurs when the valence electron concentration decreases (TiZrHfNbTa vs. VNbMoTaW). However, other factors that influence the solute-dislocation interaction energy, such as the vacancy and self-interstitial formation energies, increase. Together, they explain why the modulus-normalized yield strength of TiZrHfNbTa is greater than that of VNbMoTaW, despite the former's higher ductility. Addition of HCP elements (while maintaining the BCC structure of the alloy) makes the dislocation core structure less compact (i.e., broadens it). The HCP elements also increase the local distortion, both near the dislocation core and in the undislocated lattice far away. This effectively lowers the dislocation core energy making it easier to introduce (nucleate) dislocations, which in turn improves the ductility. Our preliminary results also suggest that the valleys in the energy landscape of TiZrHfNbTa are rather shallow, which would make dislocation motion easier in HCP-element-containing alloys (in addition to dislocation nucleation). Taken together, we respectfully disagree with the reviewer's comment that our paper doesn't demonstrate a trend. Rather, these findings provide a wealth of parameters that can be systematically varied in follow-on studies to test the effects predicted by us on strength and ductility. Of course, care must be taken to ensure that the compositional changes do not result in the formation of secondary phases (i.e., that the alloy retains its single-phase BCC structure). Within the wide compositional space of single-phase BCC solid solutions, we feel the fundamental features identified in the present study are likely to be rather universal. It is our hope that this paper will provide ample motivation for future investigations.

Line 307:

This local shuffling leads to the large lattice distortion in TiZrNbHfTa, while the electronic structure of HCP elements contributes to its low shear modulus; the latter effect has been reported before in other BCC metals [40,43]

40. Qi, L. & Chrzan, D. C. Tuning Ideal Tensile Strengths and Intrinsic Ductility of bcc Refractory Alloys, *Phys. Rev. Lett.* 112, 115503 (2014).

43. Chrzan, D. C., Sherburne, M. P., Hanlumuang, Y., Li, T. & Morris, Jr., J. W. Spreading of dislocation cores in elastically anisotropic body-centered-cubic materials: The case of gum metal, *Phys. Rev. B* 82, 184202

(2010).

Line 314:

Within the wide compositional space of single-phase BCC solid solutions, these features are likely to be universal allowing alloy designers to systematically vary the different parameters identified here to optimize strength and ductility.

Comment (2): One key point for fracture is that it is not only dictated by the mobility of dislocations, instead, it is often more critically controlled by defects, especially brittle fracture. This is one key point that in my opinion many publications on this topic are missing. This is very relevant in the context of this publication because basically all the explained phenomena should impact dislocation mobility, and thus strength but what about the fracture as a whole? I assume the authors assume that all samples do not have any “macroscopic defect”, since they are all micropillars, so what makes an alloy intrinsically more brittle? Is it only being stronger? Is there anything other than that? Because by this topic, to make an alloy ductile, we only need to make it weaker. However, without doing a deep literature survey, it would be hard to make this claim. If there is something other than strength alone, this would be very desirable to be understood. Are there alloys that can be intrinsically more ductile, so they can resist higher strengths/defects and others can?

Response to reviewer #2's comment (2):

Fracture is a “weakest link” phenomenon. As such, the various intrinsic and extrinsic factors that can lead to premature (brittle) fracture must all be suppressed (or overcome) before significant ductility can be achieved (see E. P. George and R. O. Ritchie, MRS Bull. 47, 145-150, 2022 for a recent overview of this). Intrinsic factors include, for example, sufficient mobile dislocation density, ease of dislocation motion and multiplication, and sustained work hardening. Extrinsic factors include, for example, surface cracks, flaws introduced during processing and machining, and environmental effects. Extrinsic factors are less likely to cause embrittlement if the material is intrinsically ductile; therefore, materials scientists are motivated to understand and overcome intrinsic factors that lead to brittleness.

Sticking to the alloys at hand, TiZrHfNbTa displays plastic elongations of ~8% with limited work hardening [3] and a remarkably high fracture toughness of 210 MPa m^{1/2} [X. J. Fan, R. T. Qu, Z. F. Zhang, J. Mater. Sci. Tech., 123, 70-77, 2022]. In contrast, VNbMoTaW displays zero tensile elongation (and less than 3% ductility even in compression, ref. [4]) suggesting that its resistance to crack growth (fracture toughness) is likely to be very low indeed (although no results of fracture toughness tests have been reported to date). It stands to reason, therefore, that intrinsic factors are largely responsible for these dramatic differences.

Our goal in selecting these two model alloys for investigation was to understand at least some of the responsible intrinsic factors. As summarized before, we identified several distinctive differences in the properties of these alloys, including the following. TiZrHfNbTa has a spread dislocation core while that of VNbMoTaW is compact; its shear

modulus is also lower than that of VNbMoTaW. The lattice distortion (both near the dislocation core and far away) was greater in TiZrHfNbTa than in VNbMoTaW. As a result, the dislocation core energy is lower in TiZrHfNbTa, making it easier to nucleate dislocations. Our preliminary results indicate that the valleys in the energy landscape of TiZrHfNbTa are shallower, potentially making dislocation motion (and not just dislocation nucleation) easier. We were able to link all these effects to the presence of HCP elements in TiZrHfNbTa (VNbMoTaW consists of only BCC elements).

Some of the factors identified in the present study may be rationalized by considering a simple Rice-Thomson type view of fracture [J. R. Rice, Robb Thomson, Philos. Mag. A, 29, 73-97, 1974]. In their model, a sharp cleavage crack is blunted and prevented from advancing if dislocations are spontaneously emitted from the tip of an atomically sharp crack. Blunting (ductile behavior) occurs more easily when the dislocation core is wide, the shear modulus is low, and the surface energy is high. The first two of these factors favors TiZrHfNbTa. However, the third factor, surface energy, should favor VNbMoTaW because it generally scales with the melting point (by the same token, the ideal (Griffith) fracture strength (in cleavage) of VNbMoTaW should also be higher).

However, it has been shown before [M. L. Jokl, V. Vitek, C. J. McMahon, Acta Metall. 28, 1479-1488, 1980] that fracture is not a simple either/or phenomenon as envisaged in the Rice-Thomson model; rather, plastic deformation and crack propagation often occur simultaneously. Relatively small changes in the surface (or interface energy in the case of, say, intergranular fracture) on the one hand, and shear modulus or ease of dislocation nucleation and motion on the other hand can have nonlinear multiplier effects that alter the criterion for ductile-to-brittle transition.

Given the complexity of fracture, our goal here was rather more modest than to develop a comprehensive understanding of all the factors responsible for brittle vs. ductile fracture. And in that endeavor, we believe we have succeeded. We also believe our new understanding can be applied in the development of new RHEAs that have better combinations of strength and ductility than those currently known.

Other minor points that I suggest to be addressed:

Minor comment (1): Page 5 ; please explain better the statement in lines 125-127.

Response to reviewer #2's minor comment (1):

It is widely accepted that (a) plastic deformation of BCC metals at low temperatures is controlled by screw dislocations because of their high Peierls stress, and (b) screw dislocations move by the thermally activated kink-pair nucleation and migration mechanism. In the BCC RHEAs, migration rather nucleation appears to be the rate-controlling process. Regardless, the strain rate can be described by the following equation,

$$\dot{\epsilon} = \epsilon_0 \exp\left(-\frac{H^*}{kT}\right)$$

where $\dot{\epsilon}$ is the strain rate, ϵ_0 is a pre-exponential factor, H^* is the kink-pair activation enthalpy, and k and T have their usual meanings. The empirical expression mentioned in lines 128 to 132 of our paper, $H^*=AkT$, can be derived

from the above equation as described in previous studies [21,25,A]. However, if deformation occurs by a mechanism with an activation enthalpy greater than that for kink-pair activation, the Orowan equation ($\dot{\epsilon} = \rho b v$, where ρ is mobile dislocation density, v is dislocation velocity and b is magnitude of the Burgers vector) readily indicates that the mobile dislocation density and/or dislocation velocity should rapidly increase to accommodate the higher strain rate. Therefore, a deviation from the empirical equation marks the temperature above which kink-pair nucleation and migration are not rate-controlling anymore and rapid dislocation multiplication and motion become rate controlling.

[21] D. Brunner, Comparison of Flow-Stress Measurements on High-Purity Tungsten Single Crystals with the Kink-Pair Theory, Mater. Trans. JIM. 41 (2000) 152–160.

[25] A. Giannattasio, M. Tanaka, T.D. Joseph, S.G. Roberts, An empirical correlation between temperature and activation energy for brittle-to-ductile transitions in single-phase materials, Phys. Scr. T128 (2007) 87–90.

[A] A. Seeger, L. Hollang, The Flow-Stress Asymmetry of Ultra-Pure Molybdenum Single Crystals, Mater. Trans. JIM. 41 (2000) 141–151.

As suggested, we have added the following explanation:

Line 132:

This empirical expression is valid for thermally activated processes involving kink-pair nucleation and/or migration. Therefore, when a deviation from this empirical expression occurs, it implies that kink-pair activation is no longer rate-controlling above that temperature because of a rapid increase in the mobile dislocation density [20,26-29].

Minor comment (2): Page 10 ; lines 238, 239 ; dislocation dipole is a somewhat generic terminology. Please describe in detail which type of dipole was built and also why specifically a dislocation dipole was chosen for this investigation;

also in this paragraph ; you mention that the lower nucleation energy for the loops explain the better ductility, going back to my main point number 2, would you claim this only by the strength argument or do you hypothesize that it could have any further impact?

Response to reviewer #2's minor comment (2):

(2-1) The introduction of the dislocation dipole facilitates first-principles calculations of dislocations under periodic boundary conditions. While first-principles calculations are limited to a few hundred atoms, lattice defects such as dislocations have a long-range elastic field, and this stress field becomes significant when a small model is used. From the classical theory of dislocations, it is known that this long-range stress field can be nullified by placing two pairs of dislocations with opposite signs, where each pair is a dislocation dipole. Furthermore, it has been shown that fundamental properties of screw dislocations in BCC metals can be reproduced well when a dipole is arranged as in the figure below under periodic boundary conditions [E. Clouet, Phys. Rev. Lett. 102 (2009) 055502.]. For the above reasons, we have adopted a dipole model in the present study.

Fig. Dipole configuration of dislocation pair in the periodic cell.

We have added the following sentence and reference for clarification:

Line 609:

It has been shown that fundamental properties of screw dislocations in BCC metals can be reproduced well when a dipole is arranged in the above configuration [57].

57. Clouet, E., Ventelon, L. & Willaime, F. Dislocation Core Energies and Core Fields from First Principles, Phys. Rev. Lett. 102, 055502 (2009).

(2-2) The low formation energy of dislocation implies that dislocations can form easily, allowing the dislocation density to remain relatively high even at low temperatures. Considering the fundamental relationship between dislocation motion and macroscopic plastic strain (where $\dot{\epsilon}$ is the shear strain rate, ρ the mobile dislocation density, and v the average velocity of dislocations) [E. Orowan, Proc. Phys. Soc. Lond. 52, 8 (1940)], there is a linear relationship between dislocation density and plastic strain (i.e., ductility). This is our rationale for why low formation energy of dislocations enhances ductility.

In addition, we expect the energy barrier for dislocation motion to become smaller in TiZrNbHfTa because the energy difference between the dislocation core structure and the bulk region becomes smaller, as shown in Fig. 3e. Unfortunately, due to the limitations of our computational resources it is not possible to obtain statistically valid energy barriers for dislocation motion. However, we expect the shallow valleys in the energy landscape through which dislocations move to contribute to “smoother” (easier) plastic deformation, contributing to high ductility. Again, the balance of strength and ductility/toughness can be explained by the mechanisms mentioned in response to earlier comments.

Minor comment (3): Figure 3 is a bit confusing to me, you are inverting the order, c relates to f and d to e, correct? It would be better to put each result in a separate column. Also please specify the temperature in the caption (0 Kelvin?)

Response to reviewer #2's minor comment (3):

We agree that the way the original figure was laid out was confusing. Accordingly, we have inverted the order of Fig. 3c and Fig. 3d to make these figures correspond to the order of Fig. 3e. We also now specify the temperature (0 K) in the caption of Fig. 3.

Fig. 3:

Caption of Fig. 3:

Fig. 3 | Fundamental properties, assessed at 0 K, associated with the mechanical properties of VNbMoTaW and TiZrNbHfTa. **a**, Lattice and elastic constants of VNbMoTaW and TiZrNbHfTa with ideally random configuration. **b**, Square root of the MSAD values of each constitutive element, and the average of all elements, normalized by the Burgers vector (the error bars represent standard deviations). **c-d**, Typical configuration of a screw dislocation dipole in TiZrNbHfTa (**c**) and VNbMoTaW (**d**), where the dislocation core is visualized using the differential displacement vector. **e**, Frequency distribution of the core energy normalized by the energy calculated by the elasticity theory for 135 dipole configurations in VNbMoTaW and TiZrNbHfTa (note the break in the energy scale, horizontal axis).

Reviewer #3 (Remarks to the Author):

Tsuru et al provided an interesting and detailed mechanical property study of TiZrHfNbTa and VNbMoTaW MPEAs, and binary systems to explain some underlying mechanisms controlling mechanical behavior. The manuscript is well written and easy to read. I appreciate the good amount of effort authors put in this work.

The title of this manuscript "Why some refractory high-entropy alloys are brittle and others ductile" made me very curious about knowing more about ductility in BCC refractors, which is very critical in any alloy for practical application. But the overall message changed to other mechanical properties over ductility. I really don't see that manuscript is justifying the title.

Response

As noted earlier, we have changed the title of the paper to better reflect its content.

Major concerns:

Comment (1): In Fig. 1, author didn't outline if its compressive or tensile yield stress? This important because all the experimental tests are compressive.

Response to reviewer #3's comment (1):

Thank you for pointing this out. All mechanical tests in this study were performed under compression (as originally described in the Methods section). Following the reviewer's suggestion, we have now modified the caption of Fig. 1 to make it explicitly clear.

Caption of Fig. 1:

Fig. 1 | Compressive mechanical properties of equiatomic VNbMoTaW and TiZrNbHfTa RHEAs.

Comment (2): In general, minimum of five independent slip systems are required for inducing ductility in a polycrystalline material. Although, BCC systems have too many slip systems(48) that interfere with each other and mutually obstruct slip movement, which makes it difficult to have tensile elongation. Thus, we don't expect much ductility in hcp, i.e., they are also brittle. Similarly, HCP has three slip systems (less than five slip systems), so we hardly see any ductility in HCP systems too.

In the resent manuscript, the mechanism of mixed bcc+hcp giving rise to higher ductility is still not clear to me.

Response to reviewer #3's comment (2):

We apologize for any confusion we may have inadvertently caused. As clarification, we offer the following. In the present study, BCC single-phase specimens were prepared for both VNbMoTaW and TiZrNbHfTa equiatomic alloys. Dual-phase BCC-HCP alloys were not considered. Rather, our goal was to examine the effects of HCP elements in BCC single-phase alloys. That is, while some of the individual alloying elements (Ti, Zr, Hf) are HCP at room temperature, the alloys themselves, VNbMoTaW and TiZrNbHfTa, are both BCC. Our description in the methods section was: "BCC single-phase microstructures were obtained for the TiZrNbHfTa and VNbMoTaW equiatomic alloys." To avoid confusion, we have now decided to emphasize "BCC single-phase" at the beginning of the manuscript as indicated below.

Line 93:

In the present study, BCC single-phase VNbMoTaW and TiZrNbHfTa equiatomic alloys were prepared.

Incidentally, we agree with the reviewer that the presence of an HCP phase in the originally single-phase BCC solid solution would likely make the ductility worse. To improve the ductility of HCP alloys, we believe it is important to activate $\langle c+a \rangle$ dislocations or twinning mode, including $\langle c \rangle$ component, However, that is not relevant to the present study because we are investigation BCC, not HCP, alloys. As described in the manuscript, the HCP alloying elements affect the ductility of the BCC alloys through electronic structure effects and not by the introduction of a second (distinct) HCP phase.

Comment (3): What is the compressive strain for these two alloys? It is not mentioned anywhere except in figure 2.

The results are only for 2-3% strain. How to quantify the increase in ductility in VNbMoTaW vs TiZrNbHfTa?

Response to reviewer #3's comment (3):

The difference in ductility/deformability between VNbMoTaW and TiZrNbHfTa has been reported in previous studies [3,15,16,30]. Our aim in this study was to explore the origin of this difference. That is why most compression tests were stopped at small strain levels around 2% for the ease of microstructure characterization to explore the origin of this difference. Nevertheless, for the sake of completeness, we have performed additional compression tests as described in response to the first reviewer and revised the manuscript to now include compressive stress-strain curves for VNbMoTaW and TiZrNbHfTa both below and above their respective BDTTs. As can be seen, TiZrNbHfTa is significantly more deformable than VNbMoTaW at room temperature.

[3] Senkov, O. N., Wilks, G. B., Scott, J. M. & Miracle, D. B. Mechanical properties of Nb₂₅Mo₂₅Ta₂₅W₂₅ and V₂₀Nb₂₀Mo₂₀Ta₂₀W₂₀ refractory high entropy alloys. *Intermetallics* 19, 698–706 (2011).

[15] Senkov, O. N., Gorsse, S. & Miracle, D. B. High temperature strength of refractory complex concentrated alloys, *Acta Mater.* 175, 394–405 (2019).

[16] Senkov, O. N., Scott, J. M., Senkova, S. V., Miracle, D. B., & Woodward, C. F. Microstructure and room temperature properties of a high-entropy TaNbHfZrTi alloy. *J. Alloys Compd.* 509, 6043–6048 (2011).

[30] Wang, S., Wu, M., Shu, D., Zhu, G., Wang, D. & Sun, B. Mechanical instability and tensile properties of TiZrHfNbTa high entropy alloy at cryogenic temperatures. *Acta Mater.* 201, 517–527 (2020).

Comment (4): I'm not sure I have good way of comparing different loading direction results in Fig. 2. Author's need to explain it with reason.

Response to reviewer #3's comment (4):

The reasoning behind Fig. 2 is that the angles ψ and χ in that figure give the so-called ψ - χ relation that is commonly used [31, B,C,D] to deduce the slip plane preference in BCC metals: {110}, {112}, or the maximum resolved shear stress (MRSS) plane. The activated slip plane, in turn, is closely related to the dislocation core structure, and hence is a fundamental property of plastic deformation (for example, the dislocation core structure needs to be invoked to explain why slip occurs on a certain plane, especially if it deviates from an expected one). Until our present investigation, almost nothing was known about the slip plane preference in VNbMoTaW and TiZrNbHfTa. Now, as a result of our investigations, we are able to determine that {110} and {112} are the preferred slip planes in VNbMoTaW and TiZrNbHfTa, respectively. Furthermore, we could successfully correlate these results with the DFT calculations of dislocation core structure. As for the question regarding how to determine the different loading directions, recall that the activated slip planes were determined on two orthogonal surfaces of a single grain in a polycrystalline specimen (see description of ψ - χ in the Methods section). While the loading direction for that grain can be determined from the relevant direction cosines, there is no single loading direction (Miller index) for the entire polycrystalline specimen because the different constituent grains have different orientations. That is why we confirmed what we learned from the ψ - χ relation using micropillar compression tests on single crystal specimens in which both the slip planes and the CRSS could be unambiguously determined.

[31] Yasuda, H. Y., Yamada, Y., Cho, K. & Nagase, T. Deformation behavior of HfNbTaTiZr high entropy alloy single crystals and polycrystals. *Mater. Sci. Eng. A* 809, 140983 (2021).

[B]=>44 Takeuchi, S, Furubayashi, E. & Taoka, T. Orientation dependence of yield stress in 4.4% silicon iron single crystals. *Acta Metall.* 15, 1179–1191 (1967).

[C]=>45 Lau, S. S. & Dorn, J. E. Asymmetric slip in Mo single crystals. *Phys. Status Solidi.* 2, 825–836 (1970).

[D]=>46 Takeuchi, S., Kuramoto, E. & Suzuki, T. Orientation dependence of slip in tantalum single crystals. *Acta Metall.* 20, 909–915 (1972).

We have added the following for clarification.

Line 523:

The ψ - χ relation (Fig. 2) is usually used to deduce the preference of slip plane in BCC metals: {110}, {112}, or MRSS (maximum resolved shear stress) planes [31, 44-46]. Slip plane preference is considered to be closely related to the dislocation core structure, and hence is a fundamental property of plastic deformation.

44. Takeuchi, S, Furubayashi, E. & Taoka, T. Orientation dependence of yield stress in 4.4% silicon iron single crystals. *Acta Metall.* 15, 1179–1191 (1967).

45. Lau, S. S. & Dorn, J. E. Asymmetric slip in Mo single crystals. *Phys. Status Solidi.* 2, 825–836 (1970).

46. Takeuchi, S., Kuramoto, E. & Suzuki, T. Orientation dependence of slip in tantalum single crystals. *Acta Metall.* 20, 909–915 (1972).

Comment (5): This work provides an electronic structure changes related to the ratio of HCP to BCC to control mechanical properties. Given the manuscript is about ductility, I don't see discussion or quantification of ductility in the manuscript, in terms of quantification by providing numbers?

Response to reviewer #3's comment (5):

As mentioned in response to earlier comments, although the difference in ductility/deformability between VNbMoTaW and TiZrNbHfTa has been reported in previous studies [3,15-17], for the sake of completeness, we have added a new figure showing stress-strain curves of our alloys at 77 K, room temperature, and 823 K (below and above their BDTTs). In addition, we refer to the previously measured ductilities of the two alloys as follows.

Line 85:

To develop a fundamental understanding of the key factors that control ductility and strength of RHEAs, we examined two model systems: TiZrHfNbTa ('ductile') and VNbMoTaW ('brittle'), whose strains in compression at room temperature are >35 % [16] and <3% [3], with the overarching goal of identifying broad scientific principles that can be utilized to develop a new class of structural materials for next-generation high-temperature structural applications.

Comment (6): Author's suggest for the next generation of high-temperature materials, however, I don't think without good tensile ductility such alloy can be useful for good practical application.

Response to reviewer #3's comment (6):

Admittedly, some tensile ductility is needed at room temperature even if the material is primarily intended for high-temperature applications. But it need not be extensive, just enough to withstand handling and accidental drops. For example, state-of-the-art superalloys used as cast single-crystal turbine blades typically exhibit less than 2-3% plastic elongation. The TiZrHfNbTa alloy has been shown to exhibit tensile ductility of ~8% [3], which is certainly adequate. VNbMoTaW, on the other hand, is brittle even in compression (<3% strain, [4]). However, the latter has a much higher melting point than the former and higher absolute strength, both of which are important properties for high-temperature applications. One of our goals, therefore, was to understand the origins of this ductility difference because it may help in the alloy design of stronger, higher melting alloys, with adequate tensile ductility.

Comment (7): To me, the bright-field TEM are not good enough to understand dislocation structure. Needs better TEM analysis.

Response to reviewer #3's comment (7):

To give the reader a clearer understanding of the characteristics of dislocation structures, we have now added arrows marking the edge and screw dislocations in Fig. 2 i-l.

Comment (8): Binary vs MPEAs are completely different in behavior, is it that straight forward to translate Fig. 4 results to MPEAs, this is highly over simplification.

Response to reviewer #3's comment (8):

Thank you for pointing out this important issue. We agree that the local deformation in RHEA alloys is likely more

complex than binary BCC alloys. In order to understand the trends in the preferred dislocation slip planes we have attempted to separate the effect of local distortions, due to atomic size differences, and long-range average elastic effects (i.e., the shear modulus and phonon softening) on dislocation motion. We performed first-principles calculations of the actual quinary alloys to capture the effects of local chemical variations and succeeded in capturing specific features of HEAs, such as local lattice distortion and energy distribution of dislocations (see Fig. 3.). On the other hand, regarding Fig. 2, it has been shown from preliminary studies that increasing the concentration of HCP elements changes the electronic structure which affects the preferred dislocation slip plane. In order to approximate the effect of phonon softening produced by HCP elements we use a virtual crystal approximation (VCA) on a series of binary alloys to calculate energy surface of dislocation motion within an atomistic simulation framework (Fig. 4). We believe this is a reasonable approach for estimating the effect of HCP concentration on the slip behavior observed in experiment, as it eliminates the effects of local lattice distortions produced in the HEAs while retaining an approximation of the effect of varying the HCP concentration. Using the full quinary chemistries would not allow for a simple systematic variation in HCP concentration.

For clarification, we have added the following explanation and cited related papers that discuss the generality of this approach.

Line 307:

This local shuffling leads to the large lattice distortion in TiZrNbHfTa, while the electronic structure of HCP elements contributes to its low shear modulus; the latter effect has been reported before in other BCC metals [40,43].

Line 314:

Within the wide compositional space of single-phase BCC solid solutions, these features are likely to be universal allowing alloy designers to systematically vary the different parameters identified here to optimize strength and ductility.

40. Qi, L. & Chrzan, D. C. Tuning Ideal Tensile Strengths and Intrinsic Ductility of bcc Refractory Alloys, *Phys. Rev. Lett.* 112, 115503 (2014).

43. Chrzan, D. C., Sherburne, M. P., Hanlumuang, Y., Li, T. & Morris, Jr., J. W. Spreading of dislocation cores in elastically anisotropic body-centered-cubic materials: The case of gum metal, *Phys. Rev. B* 82, 184202 (2010).

REVIEWER COMMENTS

Reviewer #1 (Remarks to the Author):

The authors have satisfactorily addressed all of my concerns. I recommend the manuscript for publication. Additionally, I believe that the previous title, "Why some refractory high-entropy alloys are brittle and others ductile," aligns well with the content of the manuscript, given its in-depth and comprehensive exploration of the fundamental origins of the ductility of RHEAs, particularly in the revised version. Nonetheless, the final decision should rightfully rest with the authors.

Reviewer #2 (Remarks to the Author):

I believe the revised manuscript version answered properly all my concerns. I believe it should be published. I feel the discussion regarding my point #2, of fracture, was very interesting and I strongly suggest adding it on the final version, even if on a supplementary materials file.

Reviewer #3 (Remarks to the Author):

Tsuru et al made extensive modifications to revised draft. Authors have included some new details in the revised draft that clarifies some issues from the first revision.

But I would surely like to elaborate on three points that still not very convincing in the current version.

(1) I'm sorry but I'm still stuck to author's original hypothesis of ductility claim.

Possibly, we all are aware about strength-ductility trade-off. Higher the yield lower the ductility and vis-a-vis. Singly phase FCC alloys are known to have higher ductility but unfortunately very low strength when compared to BCC alloys. Possibly I would agree more to the fact that adding hcp metals to bcc matrix create more charge that creates effective channels for dislocation flow along 110 slip plane of bcc, somewhat similar to fcc metals or alloys.

Moreover, the formation of secondary phases under stress provides additional mechanisms for strain accumulation, i.e. transformation-induced plasticity (TRIP) could be another possible route, which might be active and possible reason for ductility in hcp-metal-rich BCC MPEAs, which is not considered in this work or possibly I missed it?

(2) I also do not agree about low atomic displacement in VNbTaMoW (See PRMater 1, 023404 (2017); Acta Mater 257, 1-15 (2023)). If we go back to ductility idea fcc has more ductility because dislocation flow along close pack planes is easier, more they have low or no lattice distortion (see PRL 118, 205501 (2017)). Large lattice distortion will create a kind of blockage to dislocation flow, which will negatively impact ductility as found in most bcc alloys. This is opposite to what author's are claiming in TiZrNbHfTa, i.e., large distortion is responsible for higher ductility.

(3) I'm still not able to comprehend the over simplification of including binary alloys for providing mechanism underpinning ductility in five component MPEAs. I would recommend some interesting work by Samolyuk et al in PRL 126, 025501 (2021) on HCP metals vs BCC refractories. In realistic situation there can be so many interactions that may change simplified assumption (also see Mater. Res. Lett. 5, 35 (2017) where exclusion of atomic interactions lead to misinterpretation of expedients on TaWNbMo by Senokov et al published in Intermetallics 19, 698 (2011)).

It may sound repetitive but we expect more strong case, and some major advancement for publication in Nat. Commun. Although manuscript is very well written, and provides valuable and different perspective to ductility under compression.

However, I still see some contradicting hypothesis of LLD vs HCP elements tuning including oversimplified hypothesis on screw dislocation using binary alloys, which makes overall argument weaker.

Based on these I still believe that this manuscript fall short to be published in Nat. Commun.

October 15, 2023

Response To Reviewers' Comments

Ref. No: NCOMMS-23-14936-T

We thank the reviewers for their valuable feedback. Our responses are provided below, with corresponding changes highlighted in blue in the manuscript.

Reviewer #1 (Remarks to the Author):

The authors have satisfactorily addressed all of my concerns. I recommend the manuscript for publication. Additionally, I believe that the previous title, "Why some refractory high-entropy alloys are brittle and others ductile," aligns well with the content of the manuscript, given its in-depth and comprehensive exploration of the fundamental origins of the ductility of RHEAs, particularly in the revised version. Nonetheless, the final decision should rightfully rest with the authors.

Response to reviewer #1's comment:

We are gratified that all concerns of reviewer #1 were satisfactorily addressed in our revised manuscript. We are also pleased to accept the reviewer's suggestion to revert to the previous title, "Why some refractory high-entropy alloys are brittle and others ductile". We agree that the revised manuscript (when compared to the original submission) deals in depth and comprehensively with the fundamental origins of the ductility of RHEAs and thank the reviewer for pointing this out.

Line 1:

Why some refractory high-entropy alloys are brittle and others ductile

Reviewer #2 (Remarks to the Author):

I believe the revised manuscript version answered properly all my concerns. I believe it should be published. I feel the discussion regarding my point #2, of fracture, was very interesting and I strongly suggest adding it on the final version, even if on a supplementary materials file.

Response to reviewer #2's comment:

We are pleased that our revised manuscript properly addressed all of the reviewer's concerns. As suggested by the reviewer, we have added the discussion on fracture to the Supplementary Information.

Line 88:

Additional details are provided in the Supplementary Discussion.

(Supplementary Discussion and Supplementary References in Supplementary Information)

Reviewer #3 (Remarks to the Author):

Tsuru et al made extensive modifications to revised draft. Authors have included some new details in the revised draft that clarifies some issues from the first revision. But I would surely like to elaborate on three points that still not very convincing in the current version.

Comment (1): I'm sorry but I'm still stuck to author's original hypothesis of ductility claim.

Possibly, we all are aware about strength-ductility trade-off. Higher the yield lower the ductility and vis-a-vis. Singly phase FCC alloys are known to have higher ductility but unfortunately very low strength when compared to BCC alloys. Possibly I would agree more to the fact that adding hcp metals to bcc matrix create more charge that creates effective channels for dislocation flow along 110 slip plane of bcc, somewhat similar to fcc metals or alloys.

Moreover, the formation of secondary phases under stress provides additional mechanisms for strain accumulation, i.e. transformation-induced plasticity (TRIP) could be another possible route, which might be active and possible reason for ductility in hcp-metal-rich BCC MPEAs, which is not considered in this work or possibly I missed it?

Response to reviewer #3's comment (1):

Although a strength-ductility tradeoff (higher the yield, lower the ductility, and vice versa) is often observed, it is by no means universal and exceptions have been reported in single-phase FCC HEAs such as the equiatomic CrCoNi and CrMnFeCoNi alloys in which strength, ductility, and toughness all increase simultaneously with decreasing temperature instead of being traded off (e.g., Gali et al., *Intermetallics*, 39, 74-78, 2013; Otto et al., *Acta Mater.*, 61, 5743-5755, 2013; Wu et al., *Acta Mater.* 2014; Gludovatz et al., *Science*, 345, 1153-1158 2014; Liu et al., *Science*, 378, 978-983, 2022). Additionally, Liu et al. reported that the yield and ultimate strengths at 20 K of the CrCoNi FCC alloy are approximately 740 and 1290 MPa, respectively, which approach those of the TiZrHfNbTa RHEA at room temperature (this paper and references herein), while the tensile ductility of CrCoNi exceeds 40%, which is a factor of five higher than that of TiZrHfNbTa (8%). These data demonstrate that a simple inverse relationship between strength and ductility cannot adequately explain why some alloys are brittle while others are ductile; rather, in-depth analyses of dislocation core structures and energies, as undertaken here, are needed to develop fundamental understanding. That being said, it was not our intention in the present paper to compare BCC RHEAs with FCC alloys. Rather, we are comparing two BCC RHEAs, a brittle one (VNbMoTaW) and a ductile one (TiZrNbHfTa) with a goal of understanding the fundamental origins of the ductility of RHEAs (as acknowledged by reviewer 1 above). Our principal findings are as follows. The presence of HCP elements (Ti, Zr, Hf) in TiZrNbHfTa contributes to greater lattice distortion both near the dislocation core and in the lattice far away. This makes the dislocation core less compact and decreases the core energy making dislocation nucleation easier. Dislocation motion is also easier in this RHEA because of the shallower wells in the potential energy landscape, although this last finding needs additional validation in a statistically significant way. These two factors (easier dislocation nucleation and motion) contribute to the higher ductility of TiZrNbHfTa compared to VNbMoTaW.

The next sentence in comment 1 is the following: "Possibly I would agree more to the fact that adding hcp metals to bcc matrix create more charge that creates effective channels for dislocation flow along 110 slip plane of bcc,

somewhat similar to fcc metals or alloys”. We do not understand what the reviewer means by “...effective channels for dislocation flow along 110 slip plane of bcc, somewhat similar to fcc metals or alloys” because the slip plane in fcc metals and alloys is {111} not {110}. Therefore, we fail to see how the addition of hcp metals can make the slip plane in bcc RHEAs similar to that in fcc.

The last part of comment 1 deals with TRIP as a possible route for higher ductility in hcp-metal-rich RHEAs. We did in fact consider this possibility by investigating whether HCP or ω phases formed in the TiZrNbHfTa alloy which has high concentrations of HCP elements. We confirmed that no deformation-induced phase transformation occurred in the alloys considered here (TiZrNbHfTa and VNbMoTaW). That is, TRIP can be ruled out as a possible route and only mechanisms involving dislocation motion are activated in equiatomic VNbMoTaW and TiZrNbHfTa. Since dislocation-derived mechanisms contribute to the ductility of TiZrNbHfTa, we focused on that and proposed the reasons summarized above and discussed in detail in the paper.

Comment (2): I also do not agree about low atomic displacement in VNbTaMoW (See PRMater 1, 023404 (2017); Acta Mater 257, 1-15 (2023)). If we go back to ductility idea fcc has more ductility because dislocation flow along close pack planes is easier, more they have low or no lattice distortion (see PRL 118, 205501 (2017)). Large lattice distortion will create a kind of blockage to dislocation flow, which will negatively impact ductility as found in most bcc alloys. This is opposite to what author's are claiming in TiZrNbHfTa, i.e., large distortion is responsible for higher ductility.

Response to reviewer #3's comment (2):

Regarding the atomic displacement, we regret any misleading caused by our inadvertent response. To clarify, we find that the LLD of TiZrNbHfTa is larger than that of VNbTaMoW. That does not mean the atomic displacement of VNbTaMoW is small. Rather, the LLD of VNbTaMoW is comparable to that of the Cantor alloy. Our findings are consistent with those reported in one of the papers cited by the reviewer above. In PRMater 1, 023404, 2017, the lattice distortion energies (ΔE_{dis}) (defined as the difference between the energies calculated without and with atomic relaxation) of VNbTaMoW and TiZrNbHfTa were compared. The ΔE_{dis} of TiZrNbHfTa was found to be larger than that of VNbTaMoW, which implies a larger lattice relaxation in TiZrNbHfTa, which is consistent with our calculations. In Acta Mater 257, 1-15, 2023, data for TiZrNbHfTa were not provided, so a direct comparison with our results is not possible. What is clear from our results though is that the LLD of TiZrNbHfTa is larger than that of VNbTaMoW.

Regarding ductility, we don't know what else we can do at this stage but reiterate the following points that we have already made (which have not been substantially rebutted by this reviewer and have been accepted as convincing by the other two reviewers). The higher lattice distortion caused by the HCP elements in TiZrNbHfTa has the following effects. It effectively decreases the dislocation core energy (because less energy is required to create a dislocation in a more distorted lattice than in a less distorted one). This means that dislocation nucleation is easier in TiZrNbHfTa than in VNbTaMoW. Additionally, the shallower potential energy wells in TiZrNbHfTa than in VNbTaMoW imply easier dislocation motion in the former. We note that our aim was to compare two model RHEAs, one brittle and the other ductile to tease out the underlying reasons for their dramatically different mechanical

behaviors. Our findings provide ample parameters that can be systematically varied in follow-on tests to validate or disprove our findings. Our aim was not to compare FCC alloys with BCC alloys. That said, there is a glaring exception to the reviewer's sweeping claim that all BCC metals and alloys are intrinsically more brittle than FCC metals: it is well known that BCC Ta does not exhibit a ductile-brittle transition and remains ductile down to cryogenic temperatures.

Comment (3): I'm still not able to comprehend the over simplification of including binary alloys for providing mechanism underpinning ductility in five component MPEAs. I would recommend some interesting work by Samolyuk et al in PRL 126, 025501 (2021) on HCP metals vs BCC refractories. In realistic situation there can be so many interactions that may change simplified assumption (also see Mater. Res. Lett. 5, 35 (2017) where exclusion of atomic interactions lead to misinterpretation of expedients on TaWNbMo by Senokov et al published in Intermetallics 19, 698 (2011)).

It may sound repetitive but we expect more strong case, and some major advancement for publication in Nat. Commun. Although manuscript is very well written, and provides valuable and different prospective to ductility under compression.

However, I still see some contradicting hypothesis of LLD vs HCP elements tuning including oversimplified hypothesis on screw dislocation using binary alloys, which makes overall argument weaker.

Response to reviewer #3's comment (3):

We agree that deformation in RHEA alloys is likely to be more complex than in binary BCC alloys. Therefore, our approach in this paper was multifold. First, we implemented first-principles calculations of the equiatomic VNbTaMoW and TiZrNbHfTa and evaluated most of the important structural, elastic, and energetic features (such as lattice constants, elastic constants, local lattice distortion, and dislocation core structure and energy). These calculations involved the actual RHEA compositions (and not just the binaries).

However, first-principles calculations have size limitations when it comes to addressing aspects of macroscopic slip (plasticity), which are generally more tractable by MD simulations. However, classical MD simulation cannot treat these multi-component alloys due to the lack of reliable interatomic potentials for the quaternaries. An alternative is to consider the energy barrier of dislocation motion, which can provide clues to understanding slip behavior. Since direct calculation of the energy barrier for five component RHEAs is not possible due to the above reasons, we performed first-principles calculations for binary systems based on virtual crystal approximation (VCA). This is the only way to treat dislocation motion in alloy systems at this stage of first-principles calculations, and we believe it is effective in capturing salient aspects of macroscopic behavior.

The PRL paper mentioned by the reviewer has been added to the bibliography. It is worth noting, however, the last conclusion of that paper: "The significant reduction of the energy difference $E_{hcp} - E_{bcc}$ as the valence hZi goes from five toward four may be a reason for significantly improved ductility, where the formation of secondary phases under stress provides additional mechanisms for strain accumulation." As discussed earlier in Response (2), we have considered this possibility. We find that the ductility in TiZrNbHfTa is not caused by the formation of secondary phases under stress (TRIP mechanism) but by unique features of dislocation motion.

Line 304:

These transformations induce the local atomic shuffling associated with in-plane displacement along $\langle 110 \rangle$ for BCC-HCP and $\langle 111 \rangle$ for BCC- ω transitions [41-43].

43. Samolyuk, G. D., Osetsky, Y. N., Stocks, G. M. & Morris, J. R. Role of Static Displacements in Stabilizing Body Centered Cubic High Entropy Alloys. *Phys. Rev. Lett.* **126**, 025501 (2021).